# Locally Coherent Parallel Decoding in Diffusion Language Models

**Michael Hersche** [* 1]   **Nicolas Menet** [* 1 2]   **Ronan Tanios** [* 1 2]   **Abbas Rahimi** [1]

## Abstract

Diffusion language models (DLMs) have emerged as a promising alternative to autoregressive (AR) models, offering sub-linear generation latency and bidirectional capabilities that are particularly appealing for code generation and editing. Achieving sub-linear latency in discrete DLMs requires predicting multiple tokens in parallel. However, standard DLMs sample tokens independently from conditional *marginal* distributions, failing to capture the joint dependencies among concurrently generated tokens. As a result, they often lead to syntactic inconsistencies and break multi-token structures. In this work, we introduce CoDiLA (**Co**herent **Di**ffusion with **L**ocal **A**utoregression), a method that reconciles parallel sampling with local dependency modeling. Rather than forcing the DLM to resolve fine-grained syntax, CoDiLA delegates local decoding to a small, auxiliary AR model operating on the diffusion latents. This design allows for parallel generation while ensuring sequential validity within a block and maintaining core DLM capabilities, including bidirectional modeling across blocks. We demonstrate that using a highly compact auxiliary AR model (e.g., 0.6B parameters) effectively eliminates coherence artifacts, establishing a new Pareto frontier for accuracy and speed in code generation benchmarks.[1]

*Figure 1.* Our CoDiLA in action. **a)** An example of incoherent text generated by Dream-Coder-Instruct-7B in the first iteration. Due to independent modeling of marginal distributions, it predicts the incoherent token "problem" (Top-1). **b)** This work enforces local coherence using a block-wise AR model conditioned on soft local tokens. In this example, it recovers coherence by retrieving the correct token "(list" from the Top-3 candidates. Displayed prompt was simplified for illustrative purposes.

## 1. Introduction

Large language models (LLMs) have fundamentally relied on autoregressive (AR) modeling (Vaswani et al., 2017; Brown et al., 2020), generating a sequence token-by-token. While AR allows for highly parallelized training, generation remains sequential, incurring a linear latency cost with respect to the sequence length. As the demand for fast long-context generation grows, diffusion models have emerged as a compelling alternative. By formulating generation as a denoising process, diffusion models *jointly refine a set of tokens*. Crucially, when the number of denoising steps is smaller than the sequence length, the parallel refinement of diffusion models enables sub-linear generation latency, overcoming the sequential bottleneck of AR.

---
[*]Equal contribution. [1]IBM Research - Zurich [2]Department of Computer Science, ETH Zürich. Research conducted at IBM Research - Zurich. Correspondence to: Michael Hersche <michael.hersche@ibm.com>.

*Proceedings of the 43rd International Conference on Machine Learning*, Seoul, South Korea. PMLR 306, 2026. Copyright 2026 by the author(s).

[1]Code available at https://github.com/IBM/coherent-diffusion-local-autoregression

Originally applied to generative vision applications (Sohl-Dickstein et al., 2015; Ho et al., 2020; Song et al., 2020), diffusion models are also increasingly applied to natural language generation. Besides raw speed (Wang et al., 2026; Wu et al., 2026a; Ma et al., 2025) and data efficiency (Ni et al., 2025a;b; Prabhudesai et al., 2025; Rütte et al., 2026), diffusion language models (DLMs) introduce capabilities distinct from causal AR models, utilizing self-correction (Zhang et al., 2025b; Kim et al., 2025) and infilling (Zhang et al., 2025a) to solve NP-complete problems in polynomial space via backtracking (Yang et al., 2026). Such characteristics are particularly valuable in structured tasks like code generation, where non-local dependencies (e.g., library imports dictating later function calls) make the ability to backtrack and insert tokens essential. Consequently, recent DLM scaling efforts have centered on coding domains, demonstrating promise in both open-source (Gong et al., 2026; Xie et al., 2025; Song et al., 2025) and commercial (Inception et al., 2025; DeepMind, 2025) initiatives.

To adapt diffusion to the discrete nature of text, state-of-the-art (SoTA) DLMs employ an absorption state (Austin et al., 2021a; Campbell et al., 2022; Lou et al., 2024; Sahoo et al., 2024; Ou et al., 2025; Shi et al., 2024), typically represented by a dedicated [MASK] token. The forward process stochastically masks tokens to this absorption state, while the reverse process employs a bidirectional Transformer to predict the original tokens. Masked diffusion models have been successfully scaled to large parameter counts (Gong et al., 2025; Nie et al., 2025; Ye et al., 2025b; Xie et al., 2025; Bie et al., 2025; Zhu et al., 2025; Tian et al., 2025; Cheng et al., 2025). However, they struggle to predict multiple coherent tokens in parallel and thus cannot yet realize the promise of fast sub-linear generation.

**Key Challenge: Good at Global Drafting, Bad at Local Coherence.** While parallel sampling is the key to DLM efficiency, it often leads to incoherence (Liu et al., 2025a; Bansal & Sanghavi, 2025; Sun et al., 2025; Jin et al., 2025; Kang et al., 2026; Feng et al., 2025; Zhong et al., 2026). In a standard masked diffusion step, the denoising model predicts the conditional *marginal* distribution for each masked token independently, rather than the *joint* distribution across all masked tokens. By individually sampling from these univariate marginals, the model effectively assumes independence. While this assumption may hold for distant tokens, it breaks down for local structures—such as multi-token words or syntactic code blocks—resulting in incoherent outputs where individual tokens make sense in isolation but fail to form a coherent sequence (e.g., see Figure 1a).

Recent strategies enforce coherence via left-to-right generation of the sequence, either by introducing an AR verification operating on the generated sequence (Israel et al., 2025; Hu et al., 2026) or by performing self-speculative

decoding (Liu et al., 2025b). While enabling accelerated generation, many key capabilities of DLMs, such as infilling, correction, or bidirectional modeling, get lost. An alternative approach augments a DLM with an auxiliary single-layer DLM (Bansal & Sanghavi, 2025) for iterative unmasking, yet this incurs both notable accuracy degradation, due to limited modeling capacity, and overhead from repeated full-sequence attention and logit computations.

**This work: Parallel Sampling via Local Coherence.** We propose CoDiLA (**Co**herent **Di**ffusion with **L**ocal **A**utoregression), a hybrid generation paradigm that reconciles parallel sampling with local consistency (see Figure 1). This work makes the following contributions:

- We generalize discrete diffusion modeling from individual tokens to *blocks of tokens*. We theoretically prove that modeling the joint likelihood within these blocks strictly improves the achievable NELBO compared to standard conditional token-wise independence.
- Since joint modeling of large blocks is computationally intractable using a monolithic DLM, we instead adopt a lightweight auxiliary AR model. This model is *soft-conditioned* on the DLM's predicted probability distributions. Conceptually, the DLM generates a global latent draft, while the AR model executes it locally to ensure syntactic validity. Because the AR component is restricted to short, bounded blocks, CoDiLA retains the sub-linear latency benefits of diffusion.
- With a compact AR model (e.g., Qwen3-0.6B), CoDiLA establishes a new accuracy-latency Pareto frontier for code generation using Dream-Coder-Instruct-7B under static (fixed-portion) parallelism. Crucially, CoDiLA successfully preserves the base model's accuracy through dynamic (threshold-based) parallelism at $2\times$ speedup, while still supporting its native bidirectional capabilities in non-causal tasks like code infilling and planning.

## 2. Preliminaries

We consider a discrete sequence $\mathbf{x}_0 = [x_0^1, x_0^2, \ldots, x_0^L]$ of length $L$, where each token $x_0^i$ belongs to a vocabulary $\mathcal{V}$ with size $|\mathcal{V}|$. We denote the true data distribution as $q(\mathbf{x}_0)$. Let us next review the masked diffusion process that drives SoTA DLMs (Austin et al., 2021a; Campbell et al., 2022; Lou et al., 2024; Sahoo et al., 2024; Ou et al., 2025; Shi et al., 2024). We first consider the single-variable case $L = 1$, before generalizing to any sequence length $L$.

### 2.1. Univariate Discrete Diffusion

**Forward Process (Noising).** The forward process is a Markov chain that progressively corrupts the data by transitioning tokens towards a stationary noise distribution. At any timestep $t \in \{1, \ldots, T\}$, the state $x_t$ is derived from

$x_{t-1}$ through a transition matrix $\mathbf{Q}_t$ defined as $[\mathbf{Q}_t]_{jk} = q(x_t = k | x_{t-1} = j)$, parameterized by a noise schedule $\beta_t$ (Austin et al., 2021a). In masked diffusion, tokens transition to a unique absorbing state $[\texttt{MASK}] \in \mathcal{V}$. Once a token is masked, it remains masked (absorbing). Denoting with $\delta$ the Kronecker delta, the transition probability is given by:

$$q(x_t | x_{t-1}) = (1 - \beta_t)\delta_{x_t, x_{t-1}} + \beta_t \delta_{x_t, [\texttt{MASK}]}.$$

A key property of the Markovian forward process is the ability to sample $x_t$ directly from $x_0$ without iterating through intermediate steps ("skipping ahead"). By defining the cumulative transition matrix $\bar{\mathbf{Q}}_t = \mathbf{Q}_1 \mathbf{Q}_2 \ldots \mathbf{Q}_t$, the marginal distribution at timestep $t$ is given in closed form by $q(x_t = k | x_0 = j) = [\bar{\mathbf{Q}}_t]_{jk}$. This allows for efficient training by sampling arbitrary timesteps $t \sim \mathcal{U}(1, T)$ and computing the corresponding noisy states immediately. Let $\alpha_t := \prod_{s=1}^{t} 1 - \beta_s$ with $\alpha_T = 0$. Then we have

$$q(x_t | x_0) = \alpha_t \delta_{x_t, x_0} + (1 - \alpha_t)\delta_{x_t, [\texttt{MASK}]}.$$

**Reverse Process (Denoising).** The parameterized generative process learns to reverse the corruption of the forward process by approximating the intractable true posterior $q(x_{t-1} | x_t)$ with $p_\theta(x_{t-1} | x_t)$. To simplify the modeling, we parameterize $p_\theta(x_0 | x_t)$ with a neural network and propagate to $p_\theta(x_{t-1} | x_t)$ via forward process marginalization:

$$p_\theta(x_{t-1} | x_t) := \mathbb{E}_{p_\theta(x_0 | x_t)}[q(x_{t-1} | x_t, x_0)]. \qquad (1)$$

This works because $q(x_{t-1} | x_t, x_0)$, in contrast to $q(x_{t-1} | x_t)$, is fully tractable using Bayes' theorem: $q(x_{t-1} | x_t, x_0) = q(x_{t-1} | x_0)q(x_t | x_{t-1})/q(x_t | x_0)$. Therefore, to sample from $p_\theta(x_{t-1} | x_t)$, we first sample from $p_\theta(x_0 | \mathbf{x}_t)$, followed by sampling from $q(x_{t-1} | x_t, x_0)$.

### 2.2. Multivariate Discrete Diffusion

Thus far, we have considered the univariate case $L = 1$. Next, we extend Section 2.1 to sequences of arbitrary length.

**Forward Process (Noising).** We factorize the forward transition $q(\mathbf{x}_t | \mathbf{x}_{t-1})$ over the sequence positions, i.e., $q(\mathbf{x}_t | \mathbf{x}_{t-1}) = \prod_{i=1}^{L} q(x_t^i | x_{t-1}^i)$. It follows that $q(\mathbf{x}_t | \mathbf{x}_0)$ and $q(\mathbf{x}_{t-1} | \mathbf{x}_t, \mathbf{x}_0)$ are equally factorized across positions.

**Reverse Process (Denoising).** In contrast to the factorizable $q(\mathbf{x}_{t-1} | \mathbf{x}_t, \mathbf{x}_0)$, marginalization over the true data distribution $q(\mathbf{x}_0 | \mathbf{x}_t)$ breaks the factoring of $q(\mathbf{x}_{t-1} | \mathbf{x}_t)$ and hence that of the multi-step reverse transition $q(\mathbf{x}_0 | \mathbf{x}_t)$. Concurrent DLMs only partially address this: while $x_0^i$ usually depends on $x_t^j$ for all $j \in [|\mathcal{V}|]$, $p_\theta(\mathbf{x}_0 | \mathbf{x}_t)$ is typically factorized into marginals. This results in a conditional token independence bias (Kang et al., 2026; Zhong et al., 2026).

**Definition 2.1** (Conditional Token Independence). Consider a parameterized model $p_\theta(\mathbf{x}_0 | \mathbf{x}_t)$ approximating the reverse diffusion process $q(\mathbf{x}_0 | \mathbf{x}_t)$. Then the model has conditional token independence bias if $p_\theta(\mathbf{x}_0 | \mathbf{x}_t) = \prod_{i=1}^{L} p_\theta(x_0^i | \mathbf{x}_t)$.

While such factorization enables highly parallelized sampling, it introduces a coherence validity bottleneck. By sampling from univariate marginals, the model fails to capture local dependencies between tokens generated in the same step. This often results in incoherence, where individual tokens are semantically valid in isolation but fail to form coherent local structures, such as multi-token words or syntactic code blocks. The detrimental effects of incoherence are most evident with masked DLMs. To reach competitive accuracy, the number of unmasked tokens per time step is, in practice, constrained to only a few tokens.

### 2.3. Evidence Lower Bound

We train $p_\theta$ to maximize the Evidence Lower Bound (ELBO) on the data log-likelihood $\mathbb{E}_{\mathbf{x}_0 \sim q} \log p_\theta(\mathbf{x}_0)$. The negative ELBO (NELBO) decomposes into a sum of KL divergence terms between the forward process posterior and the learnable reverse process:

$$\mathcal{L}_{\text{NELBO}} := \mathbb{E}_{\mathbf{x}_0 \sim q}[\mathcal{L}_{\text{NELBO}}^{\mathbf{x}_0}] \geq \mathbb{E}_{\mathbf{x}_0 \sim q}[-\log p_\theta(\mathbf{x}_0)]$$

$$\mathcal{L}_{\text{NELBO}}^{\mathbf{x}_0} := \mathbb{E}_{q(\mathbf{x}_{1:T} | \mathbf{x}_0)} \left[ \log \frac{q(\mathbf{x}_{1:T} | \mathbf{x}_0)}{p_\theta(\mathbf{x}_{0:T})} \right] = \mathcal{L}_T + \sum_{t=1}^{T} \mathcal{L}_t$$

$$\mathcal{L}_T := \mathrm{D}_{\text{KL}}(q(\mathbf{x}_T | \mathbf{x}_0) \| p_\theta(\mathbf{x}_T))$$

$$\mathcal{L}_t := \mathbb{E}_{q(\mathbf{x}_t | \mathbf{x}_0)}[\mathrm{D}_{\text{KL}}(q(\mathbf{x}_{t-1} | \mathbf{x}_t, \mathbf{x}_0) \| p_\theta(\mathbf{x}_{t-1} | \mathbf{x}_t))].$$

Note that this decomposition requires $q(\mathbf{x}_{t-1} | \mathbf{x}_{t:T}, \mathbf{x}_0) = q(\mathbf{x}_{t-1} | \mathbf{x}_t, \mathbf{x}_0)$ and $p_\theta(\mathbf{x}_{t-1} | \mathbf{x}_{t:T}) = p_\theta(\mathbf{x}_{t-1} | \mathbf{x}_t)$. The former is satisfied under the Markovian property, and the latter is a design choice. By setting $p_\theta(\mathbf{x}_T) = \delta_{\mathbf{x}_T, [\texttt{MASK}]^{|\mathcal{V}|}}$, $\mathcal{L}_T = 0$, so $p_\theta(\mathbf{x}_{t-1} | \mathbf{x}_t)$ is only trained with loss $\sum_{t=1}^{L} \mathcal{L}_t$.

Following Sahoo et al. (2024); Shi et al. (2024); Gong et al. (2025), the parameterization of $p_\theta(\mathbf{x}_{t-1} | \mathbf{x}_t)$ in terms of $p_\theta(\mathbf{x}_0 | \mathbf{x}_t)$ enables a crucial simplification of the loss function $\mathcal{L}_t$ to a familiar cross-entropy objective (see Proposition D.1 in Appendix D):

$$\mathcal{L}_t = \mathbb{E}_{q(\mathbf{x}_t | \mathbf{x}_0)}[\mathrm{D}_{\text{KL}}(q(\mathbf{x}_{t-1} | \mathbf{x}_t, \mathbf{x}_0) \| p_\theta(\mathbf{x}_{t-1} | \mathbf{x}_t))] \qquad (2)$$

$$= \mathbb{E}_{q(\mathbf{x}_t | \mathbf{x}_0)}[\sum_{i=1}^{L} -\delta_{x_t^i, [\texttt{MASK}]} \frac{\alpha_{t-1} - \alpha_t}{1 - \alpha_t} \log p_\theta(x_0^i | \mathbf{x}_t)].$$

## 3. Method

This section presents CoDiLA (**Co**herent **Di**ffusion with **L**ocal **A**utoregression), a framework that enables parallel decoding in DLMs by enforcing local coherence (see Figure 2). We begin by partitioning the sequence of tokens into contiguous blocks, where tokens within each block are modeled jointly. We formally show that this factorization

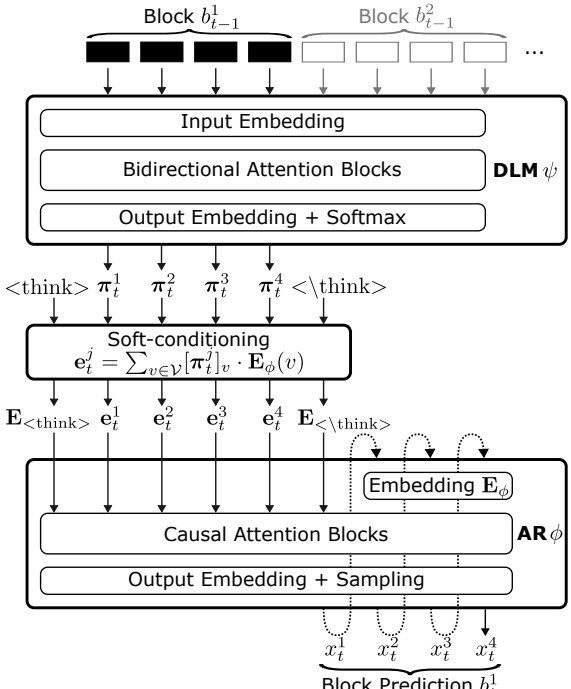

*Figure 2.* CoDiLA with a block size of $B = 4$. This example depicts the prediction of the first block ($b^1$). First, the DLM computes the token-wise conditional marginal probability vectors ($\boldsymbol{\pi}_t^j$). Next, we perform soft-conditioning by computing the expected embedding ($\mathbf{e}_t^j$) over the AR model's embedding matrix ($\mathbf{E}_\phi$), weighted by these marginals. Finally, the AR model receives these soft tokens, encapsulated by <think> and <\think> boundary tokens, to autoregressively decode a locally coherent sequence.

strictly reduces the irreducible loss (NELBO) compared to token-wise independence. Since directly modeling the joint within-block distribution becomes intractable with growing block size, we adopt a lightweight local AR model which is soft-conditioned on the DLM's marginal distributions. Conceptually, CoDiLA operates on *macro-tokens* (blocks) rather than individual tokens; as such, it retains core diffusion benefits such as self-correction and non-causal infilling.

### 3.1. Local Coherence Reduces the NELBO

To enable coherent multi-token prediction, we split the sequence $\mathbf{x}_0 = [b_0^1, b_0^2, \ldots, b_0^{L/B}]$ into blocks of tokens $b_t^i \in \mathcal{W} := \mathcal{V}^B$. Instead of applying diffusion at the token granularity, we apply it at the block granularity, modeling the joint probability as a factorization of independent blocks:

**Definition 3.1** (Conditional Block Independence)**.** Consider a parameterized model $p_\theta(\mathbf{x}_0|\mathbf{x}_t)$ approximating the reverse diffusion process $q(\mathbf{x}_0|\mathbf{x}_t)$. Then the model has conditional block independence bias if $p_\theta(\mathbf{x}_0|\mathbf{x}_t) = \prod_{i=1}^{L/B} p_\theta(b_0^i|\mathbf{x}_t)$.

In contrast to the token independence bias, the block independence bias still allows for local coherence within a

block. As shown in Theorem 3.2, this results in provable improvements on the lowest achievable NELBO.

**Theorem 3.2.** *Consider discrete diffusion on random sequences* $\mathbf{x}_0 = [b_0^1, b_0^2, \ldots, b_0^{L/B}]$ *where* $b_t^i \in \mathcal{W}$*, and a denoising model* $p_\theta$ *adopting the block independence bias of Definition 3.1. Then the smallest possible NELBO is*

$$\mathcal{B}_B := H[\mathbf{x}_0] + \sum_{t=1}^T \Big( \underbrace{\sum_{i=1}^{L/B} H[b_{t-1}^i|\mathbf{x}_t] - H[\mathbf{x}_{t-1}|\mathbf{x}_t]}_{\text{total correlation across blocks}} \Big).$$

*Further, suppose* $b_t^i = [x_t^{(i-1)\cdot B+1}, \ldots, x_t^{i\cdot B}]$ *are blocks of tokens* $x_t^k \in \mathcal{V}$*. Then,* $\mathcal{B}_1 \geq \mathcal{B}_B$ *with* $\mathcal{B}_1 - \mathcal{B}_B$ *given by*

$$\sum_{t=1}^T \sum_{i=1}^{L/B} \Big( \underbrace{\sum_{j=1}^B H[x_{t-1}^{(i-1)\cdot B+j}|\mathbf{x}_t] - H[b_{t-1}^i|\mathbf{x}_t]}_{\text{total correlation within block i}} \Big).$$

Here, $H[\mathbf{x}] := \mathbb{E}_{x\sim q}[-\log q(\mathbf{x})]$ and its conditional variants denote the entropy under the true distribution $q$. The proof is provided in Appendix D. Variants of this statement were already given by Huang et al. (2022); Liu et al. (2025a); Kang et al. (2026); Zhong et al. (2026), but we are the first to cover block sizes $B > 1$ and quantify $\mathcal{B}_1 - \mathcal{B}_B$. It follows from Theorem 3.2 that the smallest possible NELBO, and thus the irreducible modeling error, is minimized by choosing the largest possible block size. Indeed, as shown in Figure 3, our CoDiLA empirically achieves a lower loss $\mathcal{L}_t$ for larger block sizes. However, as we demonstrate next, large blocks introduce additional sequential computations.

### 3.2. Modeling the Local Joint Probability with AR

Directly modeling the distribution over the space of macro-tokens $\mathcal{W} = \mathcal{V}^B$ is intractable, as its size $|\mathcal{V}|^B$ grows exponentially with the block size $B$. For instance, with a standard vocabulary $|\mathcal{V}| \approx 150\,000$ (Qwen Team, 2025), even a small block size implies a prohibitively large readout matrix. We circumvent this issue by decomposing the problem: a bidirectional Transformer (DLM) provides global context, and a small AR model refines the local structure (see Figure 2). The bidirectional Transformer backbone, parameterized by $\psi$, operates at the token level. Consistent with standard discrete diffusion, it models the block probability as the product of conditional marginals:

$$p_\psi^{\text{DLM}}(b_0^i|\mathbf{x}_t) = \prod_{j=1}^B p_\psi^{\text{DLM}}(x_0^{(i-1)B+j}|\mathbf{x}_t).$$

To capture the dependencies ignored by the DLM backbone, we estimate the joint block probability using a small AR model parameterized by $\phi$. This model is conditioned on the DLM's local marginals. Let $\boldsymbol{\pi}_\psi^j(\mathbf{x}_t) \in \Delta^{|\mathcal{V}|-1}$ denote the marginal distribution predicted by the DLM for the $j$-th token in the block, i.e., $[\boldsymbol{\pi}_\psi^j(\mathbf{x}_t)]_v = p_\psi^{\text{DLM}}(x_0^{(i-1)B+j} = $

$v|\mathbf{x}_t)$. Then, the joint block probability is modeled as

$$p_\theta(b_0^i|\mathbf{x}_t) := p_\phi^{\text{AR}}\left(b_0^i|\pi_\psi(\mathbf{x}_t)\right)$$
$$= \prod_{j=1}^{B} p_\phi^{\text{AR}}\left(x_0^{(i-1)B+j}|x_0^{(i-1)B+<j}, \pi_\psi(\mathbf{x}_t)\right).$$

CoDiLA's total parameter set is $\theta = [\psi, \phi]$. Crucially, the AR model is only conditioned on the DLM's representations for the current block $i$ and it only predicts tokens for that block. This strictly limits the effective context length of the AR component to $B$ and ensures that the high latency of AR is only incurred over short, independent segments rather than the full sequence length $L$.

### 3.3. Soft-Conditioning as a Sufficient DLM-AR Interface

The DLM computes the marginal distributions over tokens within a block, and the AR decoder is asked to construct an appropriate joint distribution over these tokens. However, does the AR model really require the entire marginal, or does a top-1 truncated marginal distribution as in (Hu et al., 2026; Israel et al., 2025) suffice? Theorem 3.3 confirms that restricting the reconstruction tokens to top-k incurs an irreducible bias that may exclude the most likely token sequence from being generated.

**Theorem 3.3.** *Let $q$ be the true joint distribution over a block $b = (x^1, \ldots, x^B)$ with marginals $\pi = (\pi^1, \ldots, \pi^B)$. Let $\mathcal{F}(\pi)$ denote the **Fréchet class** of $\pi$, defined as the set of all valid joint distributions having marginals $\pi$.*

*Consider an autoregressive model $p_\phi^{AR}$ attempting to recover $q$ by selecting tokens from the support of $\pi$:*

1. ***Sufficiency of Soft-Conditioning:*** *If $p_\phi^{AR}$ is conditioned on the full marginals $\pi$, there exists a parameterization $\phi$ such that $p_\phi^{AR}(\,\cdot\,|\pi) = q(\,\cdot\,)$.*

2. ***Fréchet Class Restriction:*** *Let $\pi_{top\text{-}k}$ be the marginals truncated to the $k$ most likely tokens at each position. Conditioning on $\pi_{top\text{-}k}$ restricts the valid solution space to the constrained Fréchet class $\mathcal{F}(\pi_{top\text{-}k})$, strictly limiting the support of any recoverable distribution to the Cartesian product of the top-k sets.*

3. ***Exclusion of the Global Mode:*** *This restriction introduces an irreducible bias. There exist joint distributions $q$ where the global mode $b^* = \arg\max_b q(b)$ is strictly excluded from the support of the restricted class. Formally:*

$$\exists q \text{ such that } \forall q' \in \mathcal{F}(\pi_{top\text{-}k}), \quad q'(b^*) = 0 < q(b^*).$$
$$(3)$$

*Thus, high-probability coherent structures can be rendered unrecoverable solely due to marginal truncation.*

*Remark* 3.4. The sufficiency result in point (1) asserts that for any specific target copula implied by $q$, there exists a valid choice of $\phi$. In practice, $\phi$ is not chosen per instance but is learnt from data and shared across all blocks and contexts. Consequently, the AR model must either implement a fixed coupling or learn to predict the appropriate coupling solely from the input marginals $\pi$.

Theorem 3.3 shows that conditioning on the full local marginals is necessary and sufficient to reliably retrieve a coherent sequence. However, directly feeding very high-dimensional probability vectors to an AR model is computationally infeasible and would require training from scratch. To resolve this, we propose mapping these marginals into a representational space that aligns with the pretrained AR's existing knowledge. We achieve this via *soft-conditioning*, which projects the distribution onto AR's embedding space.

Formally, given marginals $\boldsymbol{\pi}_t^j$ for the $j$-th token in block $i$, we compute a soft embedding $\mathbf{e}_t^j$ as the expectation of the AR model's token embeddings $\mathbf{E}_\phi$ under this distribution:

$$\mathbf{e}_t^j = \sum_{v \in \mathcal{V}} [\boldsymbol{\pi}_t^j]_v \cdot \mathbf{E}_\phi(v). \qquad (4)$$

To ensure the interface aligns with the pretrained AR model's native representational space, we encapsulate the sequence of soft tokens within special boundary tokens, `<think>` (beginning of thought) and `<\think>` (end of thought). The AR model is thus prompted with the sequence

$$[\mathbf{E}_\phi(\texttt{<think>}), \mathbf{e}_t^1, \ldots, \mathbf{e}_t^B, \mathbf{E}_\phi(\texttt{<\think>})]$$

to autoregressively decode the coherent sequence $\mathbf{b}_0^i$.

### 3.4. Training the AR to Retrieve Coherent Blocks

Although an instruction-tuned AR model might conceivably be configured in-context to perform coherent retrieval, doing so would vastly increase the context length and thus the compute cost during inference. Instead, we train the AR model directly to minimize the cross-entropy loss $\mathcal{L}_t$ of Equation (2) within the end-to-end CoDiLA architecture. As previously noted, CoDiLA functions as a DLM that models block probabilities rather than token probabilities. Accordingly, we adjust the training objective to block-wise prediction. In other words, we adapt the forward noising process to mask entire blocks. The AR model is then trained based on $\mathcal{L}_t$ to predict the ground-truth tokens of the corresponding block conditioned on the DLM latents.

### 3.5. Generation Strategies with CoDiLA

Both the DLM and the auxiliary AR model generate predictive distributions. CoDiLA leverages these distributions to drive deterministic, confidence-based unmasking schedules.

We propose three generation modes: static parallelism, dynamic parallelism, and AR-based coherence verification. To guide these schedules, we use average conditional entropy as a proxy for uncertainty. For a partial block of size $k \leq B$, the AR model's uncertainty is defined as:

$$h_t^i(k) := \frac{1}{k} \sum_{j=1}^{k} H\big[p_\theta(x_0^{(i-1)B+j}|\mathbf{x}_t)\big].$$

An analogous entropy-based uncertainty measure can be computed at the DLM level using $p_\psi$.

**Static Parallelism (Full-Block Generation).** We unmask one full block per denoising iteration. At each step, we compute the full-block entropy $h_t^i(B)$ for all masked blocks and unmask the block with the lowest average entropy. As unrestricted global unmasking can lead to premature end-of-sequence (EOS) predictions, we employ two stabilization strategies. The first restricts the unmasking candidate pool to a local scope within 10 blocks relative to the current generation frontier. The second operates globally but applies a penalty to the entropy of all blocks where the DLM's top-1 prediction for every token in the block is the EOS token (similar to Ye et al. (2025b)). The main experiments are conducted with local scope, with ablations in Appendix B.3.

**Dynamic Parallelism (Partial-Block Generation).** Dynamic unmasking relaxes the constraint of enforcing a fixed unmasking rate per iteration. Instead, the model adjusts the number of unmasked tokens based on its confidence; i.e., we unmask the largest partial block whose entropy is below a specified threshold ($h_t^i(k) \leq \tau$). The AR model's uncertainty estimation $h_t^i(k)$ is representative for $k > 1$ tokens, validating the coherence of the generated text. Conversely, when the confidence only allows decoding a single token ($k = 1$), we fall back to the DLM's confidence, since the DLM can already guarantee coherence in non-parallel decoding. Besides confidence, single-token decoding also samples the token from DLM's distribution, following other coherence-enhancing methods (Bansal & Sanghavi, 2025). To favor the completion of partially decoded blocks, unmasked tokens are not counted toward the entropy average.

**Coherence Verification (AR-based Penalization).** Finally, we consider a generation mode where the AR model acts as a verifier rather than a direct generator. More precisely, within each block, we compare the top-1 of the DLM with the top-1 of the AR model. If they disagree, we reduce the unmasking confidence via fixed penalty. This approach seamlessly integrates into any confidence-based unmasking schedule and allows tokens to be unmasked in an arbitrary order both within and across blocks.

## 4. Experiments

### 4.1. Setup

We integrate CoDiLA into the SoTA instruct-tuned DLM for coding: Dream-Coder-Instruct-7B (Xie et al., 2025). We finetune the AR model on Ling-Coder-SFT (Codefuse & Ling Team, 2025), the same SFT dataset that was used for the DLM. We finetune a separate AR model for each block size for 32k steps, while keeping the DLM frozen. Our primary evaluation focuses on standard code generation using HumanEval (Chen et al., 2021), MBPP (Austin et al., 2021b), their plus versions (HumanEval+, MBPP+) (Liu et al., 2023), and BigCodeBench (full and hard) (Zhuo et al., 2025). Furthermore, to evaluate whether CoDiLA preserves the bidirectional, non-causal capabilities inherent to DLMs, we test on HumanEval-Infilling (Wu et al., 2025), synthetic planning tasks (ParallelBench (Kang et al., 2026), and Graph Traversal (Ye et al., 2025a)). All trainings and evaluations are run on a single NVIDIA A100 80GB GPU using Python 3.10.19, PyTorch 2.7.0, and CUDA 12.8, and bf16 precision. See Appendix A for more details.

### 4.2. Larger Block Sizes Reduce the Training Loss

To validate the theoretical insights from Section 3.1, we analyze the training loss of CoDiLA across varying block sizes ($B$). To ensure a direct comparison of modeling efficiency, we deviate from the standard stochastic masking process and employ a controlled strategy: we consistently mask contiguous segments of 32 tokens. For configurations with $B < 32$, the 32-token target is decomposed into multiple sub-blocks (e.g., four blocks of $B = 8$). As illustrated in Figure 3, the training loss rapidly converges for all block sizes. Crucially, increasing the block size consistently reduces the loss, empirically validating our theory that larger blocks capture richer local dependencies. Within this range, we observe no diminishing returns, suggesting that the model continues to benefit from the expanded joint context without saturation. Despite the smaller modeling

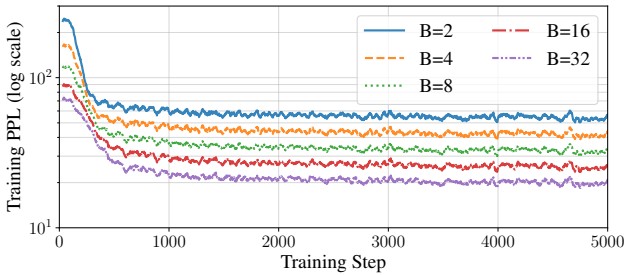

*Figure 3.* **Larger block sizes ($B$) reduce the training loss.** We compute the average perplexity weighted by the masking ratio (see Equation (2)), and display the moving average over 10 samples. The forward process always masks blocks of 32 contiguous tokens.

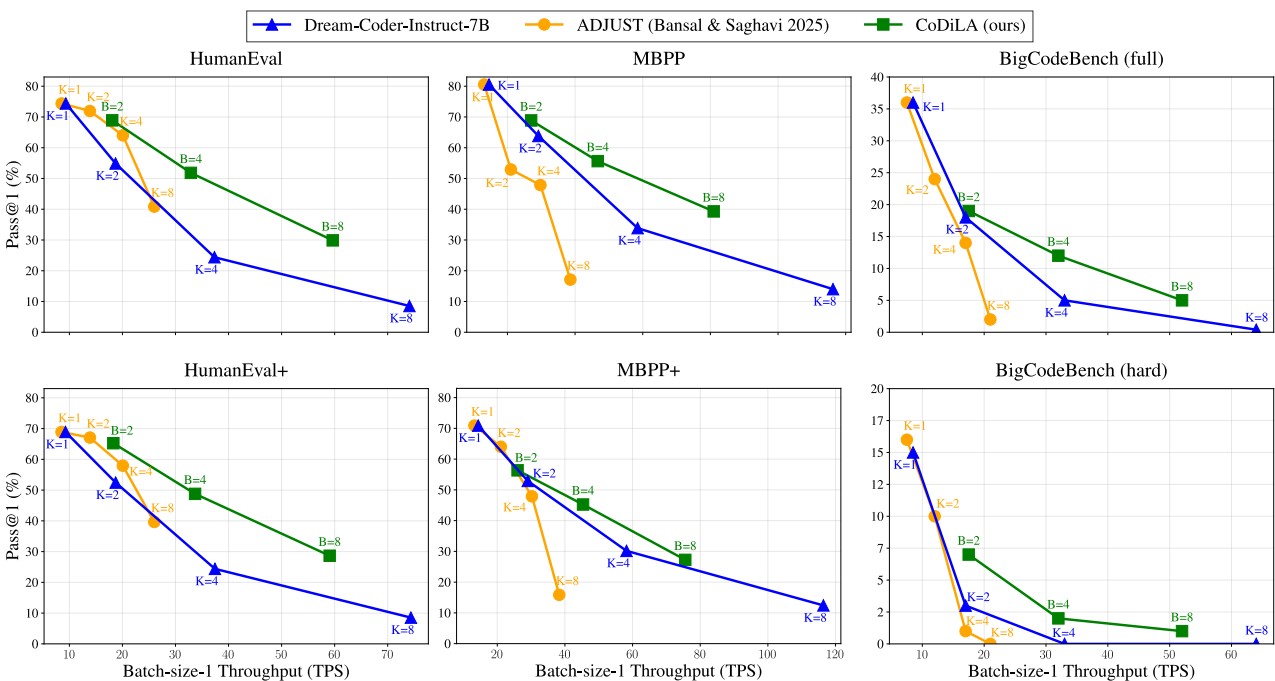

*Figure 4.* **Inference with static parallelism.** We report on Pass@1 (%) vs. Throughput (tokens/sec, batch-size 1) on a single NVIDIA A100-80GB GPU. We compare the base DLM (Xie et al., 2025), ADJUST (Bansal & Sanghavi, 2025), and our CoDiLA, all built on Dream-Coder-Instruct-7B. Parallelism is controlled by unmasking a fixed number of tokens per iteration. CoDiLA consistently achieves higher accuracy at equivalent throughput levels.

error, long sequential AR decoding in larger blocks defeat the latency benefits of parallel decoding. Hence, we set our focus on $B \leq 8$ in the inference experiments that follow.

### 4.3. Code Generation with Static Parallelism

We evaluate CoDiLA on downstream coding tasks in a *static parallelism* regime. In this setting, the model unmasks a fixed number of tokens per step. For CoDiLA with block size $B$, this entails unmasking exactly one block (the one with the highest average confidence) per iteration; consequently, the block size $B$ directly dictates the degree of parallelism. We benchmark against two baselines: (1) The standard Dream-Coder-Instruct-7B, where we enforce static parallelism by unmasking the top-$K$ tokens with lowest entropy; and (2) the ADJUST method (Bansal & Sanghavi, 2025)[2] for improved coherence, a direct competitor to CoDiLA. As shown in Figure 4, CoDiLA establishes a new Pareto optimality front across all benchmarks for Pass@1 vs. throughput with batch size 1 (a measure of latency).

In particular, CoDiLA improves accuracy compared to the Dream-Coder baseline, which we attribute to improved local coherence. We find that this gain does not simply

---

[2]The official model weights for ADJUST can be found at huggingface.co/pbansal/Dream-Coder-v0-Instruct-7B-Adjust.

stem from the AR model's standalone coding capabilities. Indeed, Qwen3-0.6B alone achieves 35% (HumanEval), 32% (HumanEval+), and 19% (both MBPP and MBPP+). CoDiLA with $B \leq 4$ outperforms Qwen3-0.6B on all benchmarks. In contrast, we observe that ADJUST fails to achieve significant speedups, likely because it does not restrict its additional coherence computation within bounded blocks.

To attribute the accuracy gains to improved coherence, we consider the reduction of syntax errors. To do so, we measure the rate at which the extraction script of the eval-harness (Gao et al., 2024) cannot extract any code. As shown in Table 1, CoDiLA reduces syntax error rate by up to 56 percentage points (pp).

*Table 1.* Syntax errors (%) on HumanEval.

| Model | K = B = 2 | K = B = 4 | K = B = 8 |
|---|---|---|---|
| Dream-Coder-Instruct-7B | 18 | 38 | 70 |
| CoDiLA | **4** | **13** | **16** |

### 4.4. Code Generation with Dynamic Parallelism

While CoDiLA achieves substantial speedup with static parallelism, accuracy degradation grows with larger block sizes—an expected trade-off for more parallelism. Here, we demonstrate how *dynamic parallelism* mitigates this by

introducing partial block unmasking based on a varying threshold $\tau$. As shown in Figure 5, CoDiLA with a block-size of $B = 4$ can bridge the gap to sequential sampling thanks to dynamic parallelism while maintaining a speed-up of $>2\times$. Notably, a larger block-size in dynamic parallelism ($B = 4$) achieves better accuracy-throughput behavior than a small block-size ($B = 2$) with static sampling.

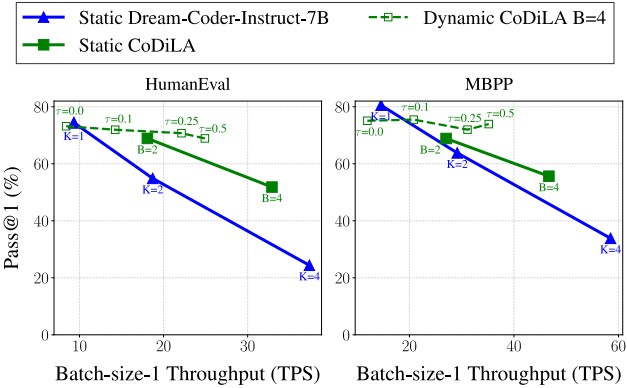

*Figure 5.* **Inference with dynamic parallelism.** We operate a dynamic CoDiLA ($B = 4$) with different entropy thresholds ($\tau$).

### 4.5. Preserving Non-Causal Capabilities

A fundamental advantage of DLMs over AR models is their bidirectional context, which enables non-causal generation such as infilling and complex planning. We demonstrate that CoDiLA can successfully preserve these global non-causal capabilities while enabling accurate parallel decoding.

**Multi-Line Code Infilling.** We implement CoDiLA on top of DreamOn (Wu et al., 2025) for the HumanEval-Infilling task. We unmask tokens predicted by the AR model with lowest mixture of entropy taken from AR and DLM. The results are shown in Table 2. We successfully accelerate generation (1.3 to 1.5 tokens/step) while maintaining the high accuracy of the sequential DLM baseline ($K = 1$), which notably outperforms pure AR models of similar scale.

*Table 2.* Multi-line code infilling on HumanEval-Infilling. [†]Baseline results are taken from Wu et al. (2025).

| | P@1 (%) | Tokens/step |
|---|---|---|
| Deepseek-Coder-6.7B[†] | 45.7 | 1 |
| Seed-Coder-8B[†] | 59.3 | 1 |
| Qwen2.5-Coder-7B[†] | 58.7 | 1 |
| DreamOn ($K = 1$) | 62.5 | 1 |
| DreamOn ($K = 2$) | 53.1 | 2 |
| DreamOn+CoDiLA ($\tau = 0.2$) | 62.5 | 1.3 |
| DreamOn+CoDiLA ($\tau = 0.5$) | 61.5 | 1.5 |

**ParallelBench** We further test CoDiLA on the waiting line tasks of ParallelBench (Kang et al., 2026). As shown in Appendix C.2, using the AR model as a verifier to penalize locally incoherent candidates during parallel sampling improves accuracy, particularly at highly parallel regimes.

**Graph Traversal** We evaluate planning capabilities on the graph traversal task from Ye et al. (2025a), shown in Appendix C.3 This experiment confirms that the auxiliary AR can be effectively trained from scratch for tasks with specialized tokenizers. As in ParallelBench, CoDiLA improves performance across highly parallel decoding regimes when using the AR model to penalize incoherent predictions.

### 4.6. Ablation Study

**Soft vs. Top-K Conditioning** We ablate the effectiveness of our soft-conditioning by comparing against Top-K-conditioning. We train variants of the AR model that conditions only on the top-K token predicted by the DLM, rather than the soft embedding used in CoDiLA. As shown in Appendix B.1, reducing the information content of the interface results in a drop in accuracy, confirming Theorem 3.3, which asserts that the AR model needs the rich signal provided by the soft tokens. Furthermore, utilizing boundary tokens (`<think>` and `<\think>`) to encapsulate these soft embeddings is crucial; removing them increases the converged NELBO from 13.6 to 15.5 for $B = 4$.

**AR Model Size** Scaling ablations in Appendix B.2 show that larger AR models with 1.7B or 4B parameters provide no consistent accuracy gain. Hence, CoDiLA's performance does not rely on a strong AR backbone.

**Candidate Scope** While the main results are based on the local scope, our ablations in Appendix B.3 confirm that the global scope with `EOS` penalty works equally well. Moreover, extending the local horizon to the full sequence length has a negligible impact on throughput (e.g., $<15\%$ when increasing from 10 to 50 blocks).

**Generation of Multiple Blocks** Appendix B.4 shows that at equivalent parallelism, generating a single larger block outperforms multiple smaller blocks (e.g., one size-8 block beats two size-4 blocks by 8 pp). This confirms that the local coherence achieved by our block structure is an essential ingredient for retaining performance in parallel decoding.

**Generation Order** To ensure our block-wise AR decoding does not impose a strict left-to-right bias on the global sequence, we analyzed CoDiLA's generation order. We computed the Spearman correlation between the AR generation and CoDiLA's sampling order on MBPP. As detailed in Appendix B.5, the correlation decreases as block size

increases, indicating that delegating local dependencies to the AR model preserves the DLM's any-order advantages.

**Throughput at Larger Batch Sizes**   While the AR model introduces a minor computational overhead at batch size 1, this cost is fully amortized at a batch size of 8, as shown in Appendix B.6. Our central results emphasize on batch size 1 to more directly measure the latency potential of DLMs.

## 5. Related Work

**Parallel Sampling Strategies.**   Dream (Ye et al., 2025b) relies on entropy-based heuristics to unmask a fixed ratio of tokens per step. To balance speed and accuracy, recent methods propose switching between exploratory (remasking) and accelerated decoding stages (Wei et al., 2026; Wang et al., 2025; Meshchaninov et al., 2025). Ma et al. (2025) introduce a hierarchical decoding strategy that divides the sequence into blocks in a divide-and-conquer fashion. Others aim to optimize the trajectory itself by learning the unmasking schedule (Bao et al., 2026), exploiting local confidence clusters (Kong et al., 2025), distilling the model via score trajectory matching (Fu et al., 2025a), or applying certainty-distillation (Chen et al., 2026). However, while these strategies enhance performance, they do not fundamentally resolve the conditional independence assumption inherent in parallel sampling. By operating on blocks, CoDiLA addresses this limitation locally and remains agnostic to the choice of scheduling or distillation strategy.

**Unordered Global Coherence Enforcement.**   Campbell et al. (2026) and Wu & Zhang (2025) propose self-speculative decoding to validate unmasked solutions in parallel, though at the cost of lower batched throughput. AD-JUST (Bansal & Sanghavi, 2025) augments a base DLM with a single-layer DLM verifier, which iteratively unmasks one token at a time, conditioned on the already decoded tokens as well as the latents of masked tokens. However, this auxiliary verifier requires training from scratch and incurs repeated global attention costs and logit computations over the full sequence. Indeed, our experiments confirm that this approach results in computational overhead and lower accuracy (see Figure 4). In contrast, CoDiLA leverages a pretrained auxiliary AR model restricted to local blocks. Our soft-conditioning avoids the overhead of full-sequence attention and eliminates the need for extensive pretraining, enabling more expressive verification with minimal cost.

**Left-to-Right Coherence Enforcement.**   APD (Israel et al., 2025) and FlashDLLM (Hu et al., 2026) employ an AR model to verify the DLM's output based on the entire generated sequence history. Similarly, TiDAR (Liu et al., 2025b) performs self-speculative decoding in a left-to-right order. These methods force the DLM into a quasi-AR mode

that effectively strips it of unique non-causal capabilities. Discrete copula diffusion (Liu et al., 2025a) combines DLM marginals with AR distributions but incurs high computational costs due to multi-pass operations over large contexts. Beyond DLMs, any-subset AR models have been utilized to accelerate sampling via any-subset speculative decoding (Guo & Ermon, 2025). However, despite the speed gains, the any-subset constraint still imposes a specialized left-to-right generation order. CoDiLA retains the DLM's global, non-autoregressive flexibility, avoiding the limitations of strict left-to-right generation.

**Efficiency Scaling and Block-Based Diffusion.**   Semi-AR block diffusion (Arriola et al., 2024; Nie et al., 2025; Arriola et al., 2025) decodes the sequence block-by-block, allowing for KV caching from previously generated segments (Liu et al., 2025c; Wu et al., 2026b). SoTA DLM frameworks, such as Fast-dLLM (Wu et al., 2026b), Fast-dLLM2 (Wu et al., 2026a), D2F (Wang et al., 2026), NBD-iff (Tian et al., 2025), Efficient-DLLM (Fu et al., 2025b), and dInfer (Ma et al., 2025), achieve competitive speeds by combining caching strategies with optimized attention. However, they typically define blocks in a fixed left-to-right order to maximize KV reuse, which again compromises the model's ability to perform arbitrary-order generation or infilling. Nevertheless, CoDiLA is compatible with these approaches, effectively introducing a second level of blocks to ensure local coherence while maintaining global efficiency.

## 6. Conclusion

We introduce CoDiLA, a hybrid framework that reconciles the parallelism of discrete DLMs with the local coherence of AR. While our theoretical analysis proves that block-based factorization strictly diminishes the irreducible modeling error compared to token-based factorization, the key enabler for practically achieving this benefit is our soft-conditioning interface to an AR model. By projecting DLM's conditional marginals into a semantic space, we create a high-capacity channel that allows a lightweight AR model to accurately retrieve coherent tokens where DLM-only inference would fail. This design effectively bridges global non-AR drafting with locally coherent execution, establishing a new Pareto front for accuracy and throughput in coding benchmarks. Future work could explore extending CoDiLA to dynamic, semantically grounded block lengths (Zhang et al., 2026).

## Impact Statement

This paper presents methodological advancements in the field of machine learning, specifically targeting the inference efficiency and accuracy of diffusion language models. As this work focuses on fundamental algorithmic improvements for parallel sampling and local coherence, we do

not foresee any specific negative societal consequences or malicious use cases that would arise directly from our contributions. The proposed techniques are general-purpose in nature and do not introduce new capabilities that would inherently facilitate harmful applications beyond the general risks already associated with large language models.

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

# A. Experimental Setting

## A.1. AR Finetuning

**Training Dataset.**   We utilize the `inclusionAI/Ling-Coder-SFT` dataset (Codefuse & Ling Team, 2025), consisting of 4.48 M instruction-response pairs, which is the same dataset specified by Xie et al. (2025) for the SFT of the Dream-Coder-7B model. We hold out a random 1% subset for validation purposes. From this filtered pool, we sample up to 300 000 training examples and 500 validation examples for our experimental runs.

**Baseline Protocol.**   Our finetuning setup is inspired by the SFT of the Dream (Ye et al., 2025b; Xie et al., 2025) and soft-masking (Hersche et al., 2026). Concretely, we employ AdamW with a cosine LR schedule, a warmup ratio of 0.03, and a max gradient norm of 7.0. We use a batch size of 1 with $8\times$ gradient accumulation (effective batch size 8). Validation is performed every 100 steps, with the best checkpoint selected by validation loss. All tokenization, chat templating, absorption `[MASK]`, and EOS/PAD handling are managed through the Dream-Coder tokenizer.

**What changes in our setup (high-level).**   We depart from the standard DLM training in three targeted ways; we detail each change in §A.1.1–A.1.3.

1. **Training roles.** The auxiliary **AR model is Qwen3-0.6B** (finetuned end-to-end), while the **DLM model (Dream-Coder-Instruct-7B) is kept frozen**.

2. **Tokenizer/interface alignment.** We use the **Dream-Coder tokenizer** for templating and masking, and perform *soft-conditioning* by multiplying the diffusion marginals with the AR embedding matrix; *we do not remap token IDs*.

3. **Masking granularity.** We replace token-wise masking with **non-overlapping block-level masking** over the response with fixed block size $B \in \{2, 4, 8\}$.

**Hardware and environment.**   All experiments are run on a single NVIDIA A100 80GB GPU using Python 3.10.19, PyTorch 2.7.0, and CUDA 12.8. We enable `bf16` and gradient checkpointing.

### A.1.1. TRAINING ROLES: AR TRAINED, DIFFUSION FROZEN

We finetune the AR model fully end-to-end and keep the DLM backbone completely frozen. Concretely:

- **Diffusion (frozen):** Dream-Coder-Instruct-7B (`Dream-org/Dream-Coder-v0-Instruct-7B`);

- **AR (trained):** Qwen3-0.6B (`Qwen/Qwen3-0.6B`), optimized with AdamW, LR $1\times10^{-5}$, cosine decay, warmup ratio 0.03, max grad norm 7.0, and weight decay 0.0.

### A.1.2. TOKENIZER AND SOFT-CONDITIONING INTERFACE

We use the Dream-Coder tokenizer for all templating, masking, `[MASK]`, EOS, and PAD handling. At masked positions, the diffusion model outputs a categorical distribution over Dream tokens; we compute the expected AR input embedding by a soft mixture over the AR embedding matrix (soft-conditioning). Importantly, we perform *no token-ID remapping*. For the model snapshots we use, token IDs `0..151664` are identical between Dream-Coder and Qwen3. Only two Dream/Qwen sentinel IDs differ:

- **ID 151665**: Dream `<|beginoftext|>` vs. Qwen `<tool_response>`,

- **ID 151666**: Dream `<|mask|>` vs. Qwen `</tool_response>`.

These mismatches are benign: Dream never predicts `<|mask|>` as a target token; and if `<|beginoftext|>` is predicted under masking, the AR side will interpret it as `<tool_response>`, which the model learns to handle. Finally, Qwen defines two additional special tokens, `<think>` and `</think>`, which we use *only* as boundary markers delimiting the soft-conditioning segment passed to the AR model. These tokens are never produced by the diffusion model, never appear in the masked loss, and therefore do not affect the DLM-AR interface.

### A.1.3. BLOCK-LEVEL MASKING

We replace token-wise masking with block-level masking over the response. The response is partitioned into non-overlapping contiguous blocks of fixed size $B \in \{2, 4, 8\}$, which serve as the atomic units of the forward noising process. For each batch, we sample $t \sim \text{Uniform}(0.2, 0.8)$ and set $p_{\text{mask}} = (1 - \varepsilon)\, t + \varepsilon$ with $\varepsilon = 10^{-3}$. Blocks are masked independently; if none is selected, we force-mask one block uniformly at random. The loss is computed only at masked positions.

### A.2. Evaluation

**Benchmarks and Metrics.** We evaluate CoDiLA in an instruction-following setting across three primary categories of code generation tasks:

- **HumanEval & MBPP**: Functional correctness is assessed on the 164 Python problems of HumanEval (Chen et al., 2021) and the sanitized version of the MBPP dataset (Austin et al., 2021b).

- **EvalPlus (HE+ & MBPP+)**: To ensure robustness against weak unit tests, we utilize the augmented test suites from Liu et al. (2023). For MBPP+, we execute the complete testing pipeline to maximize verification depth.

- **BigCodeBench**: We evaluated on both *Full* and *Hard* splits using the official v0.2.0 protocol, focusing on complex library interactions (Zhuo et al., 2025).

We utilize `lm-evaluation-harness` (v0.4.8) for HumanEval/MBPP and the `bigcodebench` (v0.2.0) framework.

**Improved Extraction Logic.** For the MBPP benchmarks, we implement a revised extraction protocol within the `lm-evaluation-harness` (Gao et al., 2024) to address known failures in the default regex-based parsing. Specifically, our pipeline ensures robust code extraction by identifying and isolating content strictly within Markdown code blocks (delimited by ```` ```python ```` and ```` ``` ````). This ensures that functional correctness is measured only on the generated implementation, mitigating the impact of surrounding conversational noise or formatting artifacts that previously led to false-negative execution failures.

**Decoding Mechanism.**

1. **Interface & Latent Projection**: Diffusion output probabilities over the Dream-Coder vocabulary are projected into the AR embedding space by computing the expected embedding relative to the AR embedding matrix. To ensure a clear termination signal, we apply a discretization step at the sequence boundary: if the diffusion model predicts the EOS token, we replace the soft mixture with the discrete (hard) embedding of the EOS token. These latents are delimited by Qwen-style `<think>` and `</think>` tags and injected into the AR model via the vLLM `EmbedsPrompt` interface.

2. **Decoding Scope**: At each denoising iteration, the model evaluates a scope of up to 10 masked blocks. For each candidate block of size $B \in \{2, 4, 8\}$, the AR model predicts its tokens. As shown in Table 5, increasing the scope results in a drop in accuracy due to premature EOS prediction. Yet, the throughput is maintained within 15%.

3. **Lowest-Entropy Unmasking**: We calculate the average per-token entropy provided by the AR executor for each candidate block. For static parallelism, the algorithm greedily unmasks the single block with the lowest average entropy, indicating the highest-confidence prediction. This yields $B$ tokens per iteration.

**Implementation Details.** All experiments are conducted in a zero-shot, single-sample ($n = 1$) setting with a fixed seed. We disable Chain-of-Thought (CoT) and external tools to isolate the model's intrinsic generation capabilities. The AR executor is configured with a temperature of 0.1 and top-$p$ of 0.8.

**Implementation.** We utilize vLLM to manage the AR model's inference. We set `gpu_memory_utilization=0.2` to ensure that both models can reside concurrently in a single device's memory. We report generation latency using CUDA-synchronized wall time, ensuring an accurate performance profile for both the diffusion and autoregressive components.

| Configuration | HumanEval/+ | MBPP/+ | BigCodeBench |
|---|---|---|---|
| Max Generation Length | 768 | 512 | 1024 |
| Temperature | 0.1 | 0.1 | 0.1 |
| Top-$p$ | 0.8 | 0.8 | 0.8 |
| Block Size $B$ | $\{2, 4, 8\}$ | $\{2, 4, 8\}$ | $\{2, 4, 8\}$ |

*Table 3.* Hyperparameter configurations across different benchmarks.

# B. Ablations

### B.1. Conditioning Strategy

We ablate the interface between the DLM and the auxiliary AR model, comparing our full soft-conditioning against Top-$K$ truncation, and evaluating the impact of explicit boundary tokens.

**Soft-Conditioning vs. Top-$K$**    As shown in Table 4, truncating the DLM's marginal distribution degrades code generation accuracy. Furthermore, computing the full expected embedding incurs negligible throughput overhead, making full soft-conditioning strictly optimal on the Pareto frontier.

**The Role of Boundary Tokens**    In our default configuration, the soft embeddings are encapsulated by explicit boundary tokens (`<think>` and `<\think>`) to align them with the AR model's discrete representational space. To quantify the importance of this design choice, we train a CoDiLA variant ($B = 4$) without these boundaries. The variant lacking boundary tokens converged to a NELBO of 15.5. In contrast, the default configuration utilizing boundary tokens achieves a much better NELBO of 13.6, confirming that explicit encapsulation is beneficial for minimizing irreducible modeling error.

*Table 4.* **Impact of conditioning strategy.** Comparison of Pass@1 (%) scores and throughput (TPS) between baseline Dream-Coder-Instruct-7B and CoDiLA with both soft- and Top-K-conditioning.

| | K=1 | | K=B=2 | | K=B=4 | | K=B=8 | |
|---|---|---|---|---|---|---|---|---|
| | P@1 | TPS | P@1 | TPS | P@1 | TPS | P@1 | TPS |
| **HumanEval** | | | | | | | | |
| Dream-Coder | 74.4 | 9 | 54.9 | 19 | 24.4 | 37 | 5.5 | 74 |
| CoDiLA (Softmax) | | | 68.9 | 18 | 51.8 | 33 | 29.9 | 60 |
| CoDiLA (Top-5) | | | 63.4 | 18 | 56.1 | 33 | 28.0 | 58 |
| CoDiLA (Top-3) | | | 61.0 | 18 | 50.0 | 31 | 31.7 | 59 |
| CoDiLA (Top-1) | | | 55.5 | 22 | 36.6 | 33 | 18.3 | 66 |
| **MBPP** | | | | | | | | |
| Dream-Coder | 80.5 | 15 | 63.8 | 29 | 33.9 | 58 | 14.0 | 116 |
| CoDiLA (Softmax) | | | 69.0 | 27 | 55.6 | 47 | 40.0 | 80 |
| CoDiLA (Top-5) | | | 62.6 | 26 | 51.4 | 47 | 33.0 | 82 |
| CoDiLA (Top-3) | | | 64.2 | 27 | 50.2 | 49 | 35.8 | 83 |
| CoDiLA (Top-1) | | | 49.4 | 26 | 33.5 | 46 | 21.7 | 73 |

### B.2. AR Model Size

We ablate the size of the AR model across HumanEval, MBPP, HumanEval+, and MBPP+. Specifically, we replace our default Qwen3-0.6B model with larger variants (1.7B and 4B parameters) to observe the impact on both generation accuracy and inference throughput.

As illustrated in Figure 6, scaling up the parameter count of the AR model does not yield any consistent or significant improvements in accuracy across any of the evaluated benchmarks. However, deploying larger AR models introduces a computational overhead. This slowdown becomes particularly pronounced when decoding larger blocks (e.g., $B = 8$), as the latency penalty of the sequential autoregressive decoding steps begins to dominate the generation time.

These results confirm that CoDiLA's improved accuracy under parallel decoding does not originate from an overly large or

highly capable AR backbone. Instead, the AR model acts as a lightweight, parameterized $n$-gram readout strictly responsible for enforcing local syntactic coherence.

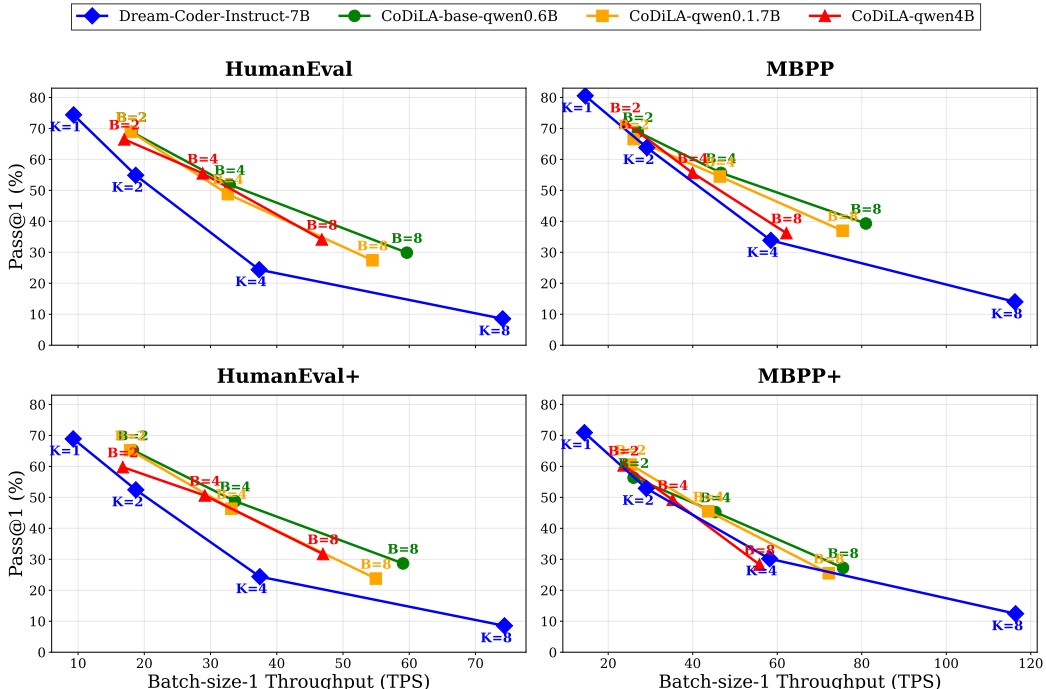

*Figure 6.* Ablation of the AR model size. CoDiLA does not require large AR models.

### B.3. Candidate Scope

**Impact of the Local Horizon**   Next, we ablate the size of the local candidate scope. As shown in Table 5, extending the lookahead horizon from 10 to 50 blocks incurs a negligible throughput penalty (less than a 15% drop in TPS). The observed accuracy degradation at larger unpenalized horizons can be attributed to premature `EOS` decoding.

*Table 5.* Increasing the lookahead window for block decoding selection has a negligible impact on the throughput (TPS), but impacts accuracy due to premature `EOS` decoding. We report CoDiLA with B=4 using static parallelism on HumanEval.

| Candidate Scope (Blocks) | Pass@1 | TPS |
|---|---|---|
| 10 (default) | 51.8% | 34.4 |
| 20 | 49.3% | 32.7 |
| 30 | 45.1% | 32.1 |
| 50 | 42.1% | 29.5 |

**Local vs. Global Strategy**   The default local scope strategy limits the candidate pool to the nearest 10 blocks relative to the current sequential generation frontier. Here, we additionally evaluate a global scope that imposes a large penalty on any block whose DLM top-1 predictions are `EOS` at all positions, thereby excluding fully uninformative blocks from the global selection objective. This prevents degenerate solutions in which empty blocks compete with content-bearing blocks during unmasking. The only exception is the leftmost candidate block, where an all-`EOS` prediction is permitted when it achieves the lowest entropy, corresponding to a termination-like decision for the sequence. As illustrated in Figure 7, both strategies successfully prevent premature `EOS` decoding and achieve comparable Pass@1 accuracy across all evaluated benchmarks.

### B.4. Generation of Multiple Blocks in Parallel

Although CoDiLA's default static decoding strategy unmasks a single block per iteration, it also permits the simultaneous decoding of multiple blocks. We ablate the effect of generating one block versus two blocks in parallel on HumanEval.

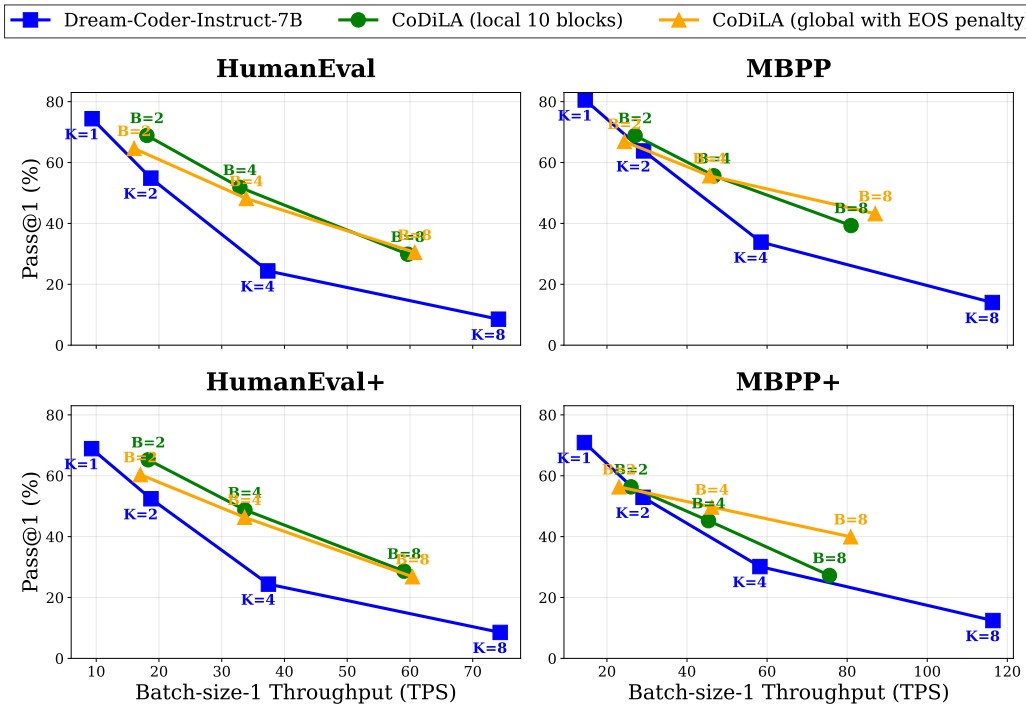

*Figure 7.* Selecting the blocks to decode globally based on DLM entropy performs comparable to a fixed lookahead window.

As detailed in Table 6, at an equivalent level of total parallelism ($K$), preserving a single contiguous block yields higher accuracy than distributing the unmasking across multiple smaller blocks.

*Table 6.* **Impact of generating multiple blocks.** Comparison of Pass@1 (%) and throughput (TPS) on HumanEval when decoding one versus two blocks in parallel. For a given block size $B$, generating two blocks doubles the total parallelism ($K = 2B$).

|  | B=2 | | B=4 | | B=8 | |
|---|---|---|---|---|---|---|
|  | Pass@1 | TPS | Pass@1 | TPS | Pass@1 | TPS |
| 1 Block ($K = B$) | 68.9 | 27 | 55.6 | 47 | 39.3 | 81 |
| 2 Blocks ($K = 2B$) | 53.3 | 54 | 31.2 | 95 | 22.3 | 159 |

### B.5. Any-Order Generation Analysis

To verify that our block-wise AR decoding does not impose a strict left-to-right bias on the global sequence, we analyzed CoDiLA's any-order capabilities. Specifically, we computed the Spearman correlation between a strictly causal left-to-right generation order and the empirical sampling order produced by the models on the MBPP benchmark.

As shown in Table 7, the base DLM exhibits a near-perfect linear correlation ($\approx 0.999$), indicating that its standard parallel decoding practically defaults to a left-to-right sequence. In contrast, CoDiLA's correlation noticeably decreases as the block size increases. This confirms that by effectively delegating local dependencies to the auxiliary AR model, CoDiLA not only preserves but actively enhances the global non-causal, any-order advantages of the underlying DLM.

### B.6. Throughput at Larger Batch Sizes

Our primary results report the throughput using a batch size of 1. To evaluate CoDiLA's efficiency in more realistic, high-throughput deployment scenarios, we also ablate the generation throughput for larger batch sizes. As shown in Table 8, moving to larger batches allows the AR model to properly utilize the GPU, effectively amortizing its computational cost. While CoDiLA exhibits a throughput penalty at batch size 1, this gap rapidly closes as the batch size increases. By batch size 8, the overhead of the AR model is completely marginalized.

*Table 7.* **Any-order generation behavior.** Spearman correlation between the model's empirical sampling order and a strict left-to-right sequence on MBPP. Lower values indicate stronger non-causal, any-order decoding behavior.

| Block Size (B = K) | Dream-Coder-7B | CoDiLA |
|---|---|---|
| 2 | 0.9998 | 0.9186 |
| 4 | 0.9999 | 0.9021 |
| 8 | 0.9994 | 0.8344 |

*Table 8.* **Throughput across varying batch sizes.** Generation speed (tokens/sec) measured on a single NVIDIA A100-80GB GPU on the HumanEval benchmark using highly parallel decoding ($K = B = 8$).

| Batch Size | Dream-Coder-7B | CoDiLA |
|---|---|---|
| 1 | 74.6 | 61.4 |
| 2 | 74.4 | 66.5 |
| 4 | 72.6 | 70.4 |
| 8 | 71.0 | 72.8 |

## C. Non-causal Tasks

### C.1. Infilling

**Setup**  We evaluate the infilling performance of CoDiLA on the multi-line HumanEval-Infilling benchmark (Fried et al., 2023). This task requires the model to synthesize missing code spans that are consistent with both preceding and succeeding multi-line context. We use the official OpenAI evaluation suite to assess functional correctness.

Our evaluation is based on the DreamOn-v0-7B checkpoint. We set the maximum prompt length to $2048$ tokens, with minimum and maximum generation lengths of $64$ and $128$ tokens, respectively. We use a batch size of $1$ and run generation for 256 decoding steps. Sampling is performed with temperature $0.2$ and top_p $0.9$.

We employ an entropy-based decoding algorithm using a threshold value $\tau \in [0.2, 0.5]$. EOS token deletion is enabled, padding to maximum sequence length is disabled, and outputs are generated in overwrite mode. Mask expansion is disabled for this baseline configuration. We evaluate on the full dataset without restricting the number of samples. Table 2 details our results.

**CoDiLA Implementation**  At each denoising iteration, we operate on contiguous masked segments, dynamically forming blocks of up to size $B = 4$.

To guide the unmasking process, we introduce a confidence signal derived from a mixture of entropy estimates from both the DLM and the auxiliary AR model. Specifically, we compute a weighted entropy score

$$H = \lambda H_{\text{DLM}} + (1 - \lambda) H_{\text{AR}},$$

where we set $\lambda = 0.5$. This symmetric mixture balances global bidirectional uncertainty from the DLM with local left-to-right fluency signals from the AR model.

Given this score, we perform confidence-based selection by sampling candidates from the AR model and unmasking tokens whose mixed entropy falls below a fixed threshold. Importantly, when a block is selected for unmasking, the final token values are always taken from the AR model, ensuring strong syntactic consistency and lexical validity, while the DLM provides global contextual guidance through the entropy signal.

To maintain robustness in fragmented or low-confidence regions, we introduce a fallback mechanism. If a block contains fewer than $B = 4$ tokens, if it includes DreamOn-specific control tokens (i.e., expansion or deletion tokens), or if its mixed entropy score exceeds the selection threshold, we revert to standard diffusion-based unmasking driven solely by the DLM.

### C.2. ParallelBench

**Setup**  We evaluate CoDiLA on the waiting line tasks of ParallelBench (Kang et al., 2026) to assess its effectiveness in preserving complex, non-causal planning capabilities during parallel generation. We follow the standard evaluation protocol

of greedy decoding with a maximum number of tokens set to 32. In this experiment, we also deploy the auxiliary AR model specifically as a *verifier* rather than a direct generator. Here, we use the Dream-v0-Instruct-7B model (no coding specialization) due better baseline performance. The AR model (Qwen3-0.6) was trained on training examples of the ParallelBench waiting line tasks to learn the syntax. To integrate CoDiLA into the entropy-based top-$K$ unmasking schedule of the base DLM, we fix the AR block size to $B = 4$. At each denoising step, the DLM computes the conditional marginals for all masked tokens. As before, for each AR block, we project these marginals into the AR model's embedding space to create soft embeddings, which are then encapsulated by `<think>` and `<\think>` boundary tokens. Then, the AR model autoregressively decodes a locally coherent sequence of tokens from these soft prompts. To implement CoDiLA verification, we now compare this AR-generated sequence against the DLM's greedy top-1 predictions for the same block. For every position where the DLM's prediction mismatches the AR model's generation—or if the AR model terminates early, leaving missing tokens—we apply a fixed penalty of $1.0$ to the DLM's base token-wise entropy. Finally, tokens are unmasked globally based on this modulated entropy. By artificially lowering the confidence of locally incoherent predictions, CoDiLA successfully deters the DLM from unmasking in parallel erroneous local structures. We evaluate across a range of unmasking rates $K \in \{1, 2, 4, 8, 16\}$, representing the total number of tokens decoded in parallel per global denoising iteration.

**Results** As shown in Table 9, utilizing CoDiLA as an entropy verifier improves the accuracy over the baseline Dream-Instruct-7B model. Notably, the benefits of the AR verifier become most pronounced at highly parallel decoding regimes where standard entropy heuristics degrade. For instance, at $K = 8$ and $K = 16$, CoDiLA improves performance by 10.5 and 7.0 percentage points, respectively. This demonstrates that penalizing local incoherence effectively stabilizes the parallel sampling trajectories, allowing the model to retain planning accuracy while decoding multiple tokens simultaneously.

*Table 9.* **ParallelBench Waiting Line.** Comparison of accuracy (%) between Dream-Instruct-7B and CoDiLA acting as a verifier.

|  | K=1 | K=2 | K=4 | K=8 | K=16 |
|---|---|---|---|---|---|
| Dream-Instruct-7B | 85.1 | **85.5** | 63.2 | 52.6 | 43.3 |
| CoDiLA | **88.6** | 85.3 | **68.6** | **63.1** | **50.3** |

## C.3. Graph Traversal (Planning)

**Task Description** The objective of the task is to navigate a graph by identifying the correct path between a specified start and goal node amid distracting, dead-end edge connections. The model is prompted with a shuffled list of all available edges, followed by the specific start and goal nodes. For example, the input format is structured as:

```
4,7 | 5,8 | 7,0 | 3,1 | ...  | 0,2 / 3,9
```

where node 3 is the start and node 9 is the goal. The model must process this unordered graph topology to output the correct, ordered sequence of edges comprising the valid path:

```
3,1 | 1,7 | 7,5 | 5,8 | 8,9
```

**Setup** We first train the baseline DLM (MGDM), which consists of a 3-layer Transformer with a total of 6M parameters, for 1200 epochs. Because this baseline utilizes a task-specific tokenizer, we train the auxiliary CoDiLA AR model from scratch for 100 epochs, adopting an identical 3-layer Transformer architecture with causal attention. During inference, we apply our coherence verification strategy: we use the standard top-$K$ decoding provided by MGDM (which relies on maximum probability instead of entropy for confidence scoring) and actively re-mask any mismatched tokens between the DLM predictions and the AR verification step. The maximum generation length (including the prompt) is set to 80 both for training and testing.

**Results** As shown in Table 10, CoDiLA successfully improves planning accuracy, with the highest gains observed at larger block sizes and fewer decoding iterations (i.e., highly parallel regimes). Because the AR model in this configuration acts as a local verifier rather than a direct sequence generator, expanding the block size provides richer joint context and strictly improves performance.

*Table 10.* Accuracy (%) on graph traversal (planning)

| Decoding iterations | MGDM (repr.) | CoDiLA ($B = 4$) | CoDiLA ($B = 8$) |
|---|---|---|---|
| 2 | 94.4 | 95.3 | **95.4** |
| 4 | 96.5 | 96.8 | **96.9** |
| 8 | 97.8 | **98.2** | **98.2** |
| 16 | 99.2 | **99.3** | **99.3** |
| 20 | 99.3 | **99.5** | 99.4 |

# D. Proofs

**Theorem** (Restatement of Theorem 3.2). *Consider discrete diffusion on random sequences* $\mathbf{x}_0 = [b_0^1, b_0^2, \ldots, b_0^{L/B}]$ *where* $b_t^i \in \mathcal{W}$, *and a denoising model* $p_\theta$ *adopting the block independence bias of Definition 3.1. Then the smallest possible NELBO is*

$$\mathcal{B}_B := H[\mathbf{x}_0] + \sum_{t=1}^{T} \Big( \underbrace{\sum_{i=1}^{L/B} H[b_{t-1}^i|\mathbf{x}_t] - H[\mathbf{x}_{t-1}|\mathbf{x}_t]}_{\textit{total correlation across blocks}} \Big).$$

*Further, suppose* $b_t^i = [x_t^{(i-1)\cdot B+1}, \ldots, x_t^{i \cdot B}]$ *are blocks of tokens* $x_t^k \in \mathcal{V}$. *Then,* $\mathcal{B}_1 \geq \mathcal{B}_B$ *with* $\mathcal{B}_1 - \mathcal{B}_B$ *given by*

$$\sum_{t=1}^{T} \sum_{i=1}^{L/B} \Big( \underbrace{\sum_{j=1}^{B} H[x_{t-1}^{(i-1)\cdot B+j}|\mathbf{x}_t] - H[b_{t-1}^i|\mathbf{x}_t]}_{\textit{total correlation within block } i} \Big).$$

*Proof.* Following Proposition 1 in Liu et al. (2025a), we first derive the closed-form solution to the negative ELBO for a model only constrained by the structural independence bias (Definition 2.1), but at the granularity of blocks instead of tokens (Definition 3.1). Then, we demonstrate and quantify the decrease in the smallest achievable NELBO as the block size $B$ increases.

*1. Decomposition of the factorization gap*

Following Ho et al. (2020); Sohl-Dickstein et al. (2015), the negative ELBO $\mathcal{L}_{\text{NELBO}}$ can be decomposed as follows:

$$\mathcal{L}_{\text{NELBO}} = \mathbb{E}_{\mathbf{x}_0 \sim q}[\mathcal{L}_{\text{NELBO}}^{x_0}] = \mathbb{E}_{\mathbf{x}_{0:T} \sim q}\Big[ \log \frac{q(\mathbf{x}_{1:T}|\mathbf{x}_0)}{p_\theta(\mathbf{x}_{0:T})} \Big] = H[\mathbf{x}_0] + \mathbb{E}_{\mathbf{x}_{0:T} \sim q}\Big[ \log \frac{q(\mathbf{x}_{0:T})}{p_\theta(\mathbf{x}_{0:T})} \Big]$$

$$= H[\mathbf{x}_0] + \mathbb{E}_{\mathbf{x}_{0:T} \sim q}\Big[ \log \frac{q(\mathbf{x}_T)}{p_\theta(\mathbf{x}_T)} + \sum_{t=1}^{T} \log \frac{q(\mathbf{x}_{t-1}|\mathbf{x}_t)}{p_\theta(\mathbf{x}_{t-1}|\mathbf{x}_t)} \Big]$$

$$= H[\mathbf{x}_0] + \text{D}_{\text{KL}}(q(\mathbf{x}_T) \| p_\theta(\mathbf{x}_T)) + \sum_{t=1}^{T} \mathbb{E}_{\mathbf{x}_t \sim q}\big[ \text{D}_{\text{KL}}(q(\mathbf{x}_{t-1}|\mathbf{x}_t) \| p_\theta(\mathbf{x}_{t-1}|\mathbf{x}_t)) \big]. \quad (5)$$

Note the use of Markovianity of both the true noising process $q$ and the learned denoising process $p_\theta$. Since conditional independence (Definition 3.1) does not impose restrictions on $p_\theta(\mathbf{x}_T)$, we set $p_\theta(\mathbf{x}_T) = q(\mathbf{x}_T)$ ensuring that $\text{D}_{\text{KL}}(q(\mathbf{x}_T) \| p_\theta(\mathbf{x}_T)) = 0$. Note that this is achievable in practice, since $q(\mathbf{x}_T)$ converges to a known stationary distribution as $T \to \infty$. Therefore, we must only derive the lowest achievable value of

$$\mathbb{E}_{\mathbf{x}_t \sim q}\big[ \text{D}_{\text{KL}}(q(\mathbf{x}_{t-1}|\mathbf{x}_t) \| p_\theta(\mathbf{x}_{t-1}|\mathbf{x}_t)) = \mathbb{E}_{\mathbf{x}_t \sim q}\big[ \text{D}_{\text{KL}}(q(\mathbf{x}_{t-1}|\mathbf{x}_t) \| \prod_{i=1}^{L/B} p_\theta(b_{t-1}^i|\mathbf{x}_t))$$

$$= \mathbb{E}_{\mathbf{x}_{t-1:t} \sim q}[-\log \prod_{i=1}^{L/B} p_\theta(b_{t-1}^i|\mathbf{x}_t)] - H[\mathbf{x}_{t-1}|\mathbf{x}_t]$$

$$= \mathbb{E}_{\mathbf{x}_t \sim q}\Big[ \sum_{i=1}^{L/B} \underbrace{\mathbb{E}_{b_{t-1}^i \sim q(b_{t-1}^i|\mathbf{x}_t)}[-\log p_\theta(b_{t-1}^i|\mathbf{x}_t)]}_{\text{cross entropy of } p_\theta(b_{t-1}^i|\mathbf{x}_t) \text{ with respect to } q(b_{t-1}^i|\mathbf{x}_t)} \Big] - H[\mathbf{x}_{t-1}|\mathbf{x}_t].$$

Since the cross-entropy factorizes across blocks, and because the cross-entropy is minimized if the distributions coincide, we may set $p_\theta(b_{t-1}^i|\mathbf{x}_t) = q(b_{t-1}^i|\mathbf{x}_t)$. Given this choice for $p_\theta$, we arrive at the following expression:

$$\mathbb{E}_{\mathbf{x}_t \sim q}\big[ \text{D}_{\text{KL}}(q(\mathbf{x}_{t-1}|\mathbf{x}_t) \| p_\theta(\mathbf{x}_{t-1}|\mathbf{x}_t)) = \sum_{i=1}^{L/B} H[b_{t-1}^i|\mathbf{x}_t] - H[\mathbf{x}_{t-1}|\mathbf{x}_t].$$

Plugging the result into Equation (5), we obtain the desired expression for $\mathcal{B}_B$, the lowest achievable NELBO.

*2. Decrease in NELBO for non-trivial block sizes*

We now quantify the reduction in the lower bound $\mathcal{B}_B$ relative to the token-level baseline $\mathcal{B}_1$. Recall that a block $b_{t-1}^i$ is composed of the subsequence of tokens $[x_{t-1}^{(i-1)\cdot B+1}, \ldots, x_{t-1}^{i\cdot B}]$. Subtracting the expression for $\mathcal{B}_B$ derived in Part 1 from the expression for $\mathcal{B}_1$ (where block size is 1), the terms $H[\mathbf{x}_0]$ and $\sum_{t=1}^{T} H[\mathbf{x}_{t-1}|\mathbf{x}_t]$ cancel out, yielding:

$$\mathcal{B}_1 - \mathcal{B}_B = \sum_{t=1}^{T} \left( \sum_{k=1}^{L} H[x_{t-1}^k|\mathbf{x}_t] - \sum_{i=1}^{L/B} H[b_{t-1}^i|\mathbf{x}_t] \right).$$

By grouping the token-level entropies according to their block assignment, we can rewrite the first sum as a nested summation over blocks $i$ and positions $j$ within that block:

$$\sum_{k=1}^{L} H[x_{t-1}^k|\mathbf{x}_t] = \sum_{i=1}^{L/B} \sum_{j=1}^{B} H[x_{t-1}^{(i-1)\cdot B+j}|\mathbf{x}_t].$$

Substituting this back into the difference equation allows us to merge the sums over $i$, finishing the proof:

$$\mathcal{B}_1 - \mathcal{B}_B = \sum_{t=1}^{T} \sum_{i=1}^{L/B} \left( \sum_{j=1}^{B} H[x_{t-1}^{(i-1)\cdot B+j}|\mathbf{x}_t] - H[b_{t-1}^i|\mathbf{x}_t] \right).$$

The term within the brackets corresponds precisely to the total correlation (or multivariate mutual information) within block $i$. Since total correlation is non-negative (the entropy of a joint variable is always less than or equal to the sum of its marginal entropies), it follows that $\mathcal{B}_1 - \mathcal{B}_B \geq 0$. Thus, a non-trivial block size strictly decreases the lower bound on the NELBO by internalizing the dependencies that are local to the block. $\qquad\square$

**Theorem** (Restatement of Theorem 3.3). *Let $q$ be the true joint distribution over a block $b = (x^1, \ldots, x^B)$ with marginals $\pi = (\pi^1, \ldots, \pi^B)$. Let $\mathcal{F}(\pi)$ denote the **Fréchet class** of $\pi$, defined as the set of all valid joint distributions having marginals $\pi$. Consider an autoregressive model $p_\phi^{AR}$ attempting to recover $q$ by selecting tokens from the support of $\pi$:*

1. ***Sufficiency of Soft-Conditioning:** If $p_\phi^{AR}$ is conditioned on the full marginals $\pi$, there exists a parameterization $\phi$ such that $p_\phi^{AR}(\,\cdot\,|\pi) = q(\,\cdot\,)$.*

2. ***Fréchet Class Restriction:** Let $\pi_{top\text{-}k}$ be the marginals truncated to the $k$ most likely tokens at each position. Conditioning on $\pi_{top\text{-}k}$ restricts the valid solution space to the constrained Fréchet class $\mathcal{F}(\pi_{top\text{-}k})$, strictly limiting the support of any recoverable distribution to the Cartesian product of the top-$k$ sets.*

3. ***Exclusion of the Global Mode:** This restriction introduces an irreducible bias. There exist joint distributions $q$ where the global mode $b^* = \arg\max_b q(b)$ is strictly excluded from the support of the restricted class. Formally:*

$$\exists q \text{ such that } \forall q' \in \mathcal{F}(\pi_{top\text{-}k}), \quad q'(b^*) = 0 < q(b^*). \tag{6}$$

*Thus, high-probability coherent structures can be rendered unrecoverable solely due to marginal truncation.*

*Proof.* *1. Sufficiency.* By Sklar's Theorem, any discrete joint distribution $q$ uniquely decomposes into its marginals $\pi$ and a copula $C$. Since $\pi$ defines the input domain and $p_\phi^{AR}$ acts as a universal function approximator, there exists a parameter setting $\phi$ that implements the copula $C$, recovering $q$ exactly.

*2. Fréchet Class Restriction.* Let $\mathcal{S}_k^i = \{t \in \mathcal{V}|\text{rank}(t, \pi^i) \leq k\}$ be the set of top-$k$ tokens at position $i$. Define the truncated marginal distribution $\pi_{top\text{-}k}^i(t) \propto \pi(t) \cdot \mathbb{1}_{t \in S_k^i}$ Any distribution $q' \in \mathcal{F}(\pi_{top\text{-}k})$ must satisfy the marginal constraints of $\pi_{top\text{-}k}$. Since the probability mass of tokens outside $\mathcal{S}_k^i$ is effectively zeroed out in the input, any valid joint distribution $q'$ must have its support contained within the Cartesian product of these sets:

$$\text{supp}(q') \subseteq \mathcal{S}_k^1 \times \mathcal{S}_k^2 \times \cdots \times \mathcal{S}_k^B. \tag{7}$$

*3. Exclusion of the Global Mode.* We demonstrate this exclusion via a counter-example. Let $B = 2$, $k = 1$, and $\mathcal{V} = \{Roger, Houston, You, I, They\}$. Consider a distribution $q(x^1, x^2)$ with the following probability mass function:

- $q(Roger, Roger) = 0.45$ (The coherent mode $b^*$).

- $q(Houston, You) = 0.25$.

- $q(Houston, I) = 0.25$.

- $q(Houston, They) = 0.05$.

The induced marginals are:

- Position 1: $\pi^1(Roger) = 0.45$, $\pi^1(Houston) = 0.55$. The top-1 token is $Houston$, so $\mathcal{S}_1^1 = \{Houston\}$.

- Position 2: $\pi^2(Roger) = 0.45$, $\pi^2(You) = 0.25$, $\pi^2(I) = 0.25$, $\pi^2(They) = 0.05$. The top-1 token is $Roger$, so $\mathcal{S}_1^2 = \{Roger\}$.

The restricted support is $\mathcal{S}_1^1 \times \mathcal{S}_1^2 = \{(Houston, Roger)\}$. However, the true global mode is $b^* = (Roger, Roger)$. Since $Roger \notin \mathcal{S}_1^1$, the true mode lies outside the restricted support. Consequently, $q'(Roger, Roger) = 0$ for all $q' \in \mathcal{F}(\pi_{top-1})$, despite $(Roger, Roger)$ being the single most likely sequence. This proves that top-$k$ truncation prevents the recovery of the true mode in the general case. $\qquad\square$

**Proposition D.1** (Closed-Form KL-Divergence for Masked Diffusion according to Sahoo et al. (2024); Shi et al. (2024); Gong et al. (2025)). *Consider masked diffusion, i.e., a Markov chain $\mathbf{x}_0, \ldots, \mathbf{x}_T$ with position-wise independent forward process $q(\mathbf{x}_t|\mathbf{x}_{t-1}) = \prod_{i=1}^L q(x_t^i|x_{t-1}^i)$, where $q(x_t^i|x_{t-1}^i) = (1-\beta_t)\delta_{x_t^i, x_{t-1}^i} + \beta_t \delta_{x_t^i, [MASK]}$. Define $\alpha_t := \prod_{s=1}^t 1 - \beta_s$ and let $p_\theta(\mathbf{x}_{t-1}|\mathbf{x}_t) = \prod_{i=1}^L p_\theta(x_{t-1}^i|\mathbf{x}_t)$ where $p_\theta(x_{t-1}^i|\mathbf{x}_t) = \mathbb{E}_{p_\theta(x_0^i|\mathbf{x}_t)}[q(x_{t-1}^i|x_t^i, x_0^i)]$, i.e., the model $p_\theta$ is assumed to have conditional token independence bias (Definition 2.1). Assume further that $p_\theta(x_0^i|\mathbf{x}_t) = \delta_{x_0^i, x_t^i}$ if $x_t^i \neq [MASK]$ and $p_\theta(x_0^i = [MASK]|\mathbf{x}_t) = 0$. Then*

$$D_{KL}(q(\mathbf{x}_{t-1}|\mathbf{x}_t, \mathbf{x}_0)||p_\theta(\mathbf{x}_{t-1}|\mathbf{x}_t)) = \sum_{i=1}^L -\delta_{x_t^i = [MASK]} \frac{\alpha_{t-1} - \alpha_t}{1 - \alpha_t} \log p_\theta(x_0^i|\mathbf{x}_t).$$

*Proof.* We follow Sahoo et al. (2024); Shi et al. (2024); Gong et al. (2025). First, note that since $q(x_t^i|x_0^i) = \alpha_t \delta_{x_t^i, x_0^i} + (1-\alpha_t)\delta_{x_t^i, [MASK]}$, a direct application of Bayes' theorem results in the closed-form expression

$$q(x_{t-1}^i|x_t^i, x_0^i) = \begin{cases} 1 & \text{if } x_{t-1}^i = x_t^i = x_0^i, \\ \frac{1-\alpha_{t-1}}{1-\alpha_t} & \text{if } x_{t-1}^i = x_t^i = [MASK], \\ \frac{\alpha_{t-1}-\alpha_t}{1-\alpha_t} & \text{if } x_{t-1}^i = x_0^i, x_t^i = [MASK], \\ 0 & \text{otherwise} \end{cases}.$$

We introduce the abbreviation $D$ and apply position-wise independence of $q(\mathbf{x}_{t-1}|\mathbf{x}_t, \mathbf{x}_0)$ and $p_\theta(\mathbf{x}_{t-1}|\mathbf{x}_t)$:

$$D := D_{KL}(q(\mathbf{x}_{t-1}|\mathbf{x}_t, \mathbf{x}_0)||p_\theta(\mathbf{x}_{t-1}|\mathbf{x}_t)) = \mathbb{E}_{q(\mathbf{x}_{t-1}|\mathbf{x}_t, \mathbf{x}_0)}\left[\frac{\log q(\mathbf{x}_{t-1}|\mathbf{x}_t, \mathbf{x}_0)}{\log p_\theta(\mathbf{x}_{t-1}|\mathbf{x}_t)}\right] = \sum_{i=1}^L \mathbb{E}_{q(x_{t-1}^i|x_t^i, x_0^i)}\left[\frac{\log q(x_{t-1}^i|x_t^i, x_0^i)}{\log p_\theta(x_{t-1}^i|\mathbf{x}_t)}\right].$$

Next, we use that if $x_t^i \neq [MASK]$, then $x_t^i = x_0^i$ and hence $q(x_{t-1}^i|x_t^i, x_0^i) = p_\theta(x_{t-1}|\mathbf{x}_t) = \delta_{x_{t-1}^i, x_0^i}$. Hence, we get

$$D_{KL}(q(\mathbf{x}_{t-1}|\mathbf{x}_t, \mathbf{x}_0)||p_\theta(\mathbf{x}_{t-1}|\mathbf{x}_t)) = \sum_{i=1}^L \delta_{x_t^i = [MASK]} D_{KL}(q(x_{t-1}^i|x_t^i = [MASK], x_0^i)||p_\theta(x_{t-1}^i|\mathbf{x}_t)).$$

Now, note that for $x_{t-1}^i = [MASK]$, the probability $q(x_{t-1}^i|x_t^i, x_0^i)$ does not depend on $x_0^i$, resulting in $p_\theta(x_{t-1}^i|\mathbf{x}_t) = q(x_{t-1}^i|x_t^i, x_0^i) \,\forall x_0^i$. Therefore, the only non-vanishing additive term in the KL divergence occurs when $x_{t-1}^i = x_0^i$, i.e.,

$$D = \sum_{i=1}^L \delta_{x_t^i = [MASK]} q(x_{t-1}^i = x_0^i|x_t^i = [MASK], x_0^i) \log \frac{q(x_{t-1}^i = x_0^i|x_t^i = [MASK], x_0^i)}{p_\theta(x_{t-1}^i = x_0^i|\mathbf{x}_t)}.$$

Finally, we compute $p_\theta(x_{t-1}^i = x_0^i | \mathbf{x}_t)$ provided that $x_t^i = \texttt{[MASK]}$. First, note that $q(x_{t-1}^i = x_0^i | x_t^i = \texttt{[MASK]}, \tilde{x}_0^i) = \frac{\alpha_{t-1} - \alpha_t}{1 - \alpha_t} \delta_{x_0^i, \tilde{x}_0^i} + \frac{1 - \alpha_{t-1}}{1 - \alpha_t} \delta_{x_0^i, \texttt{[MASK]}}$. Then, due to the assumption that $p_\theta(x_0^i = \texttt{[MASK]} | \mathbf{x}_t) = 0$, it follows that

$$p_\theta(x_{t-1}^i = x_0^i | \mathbf{x}_t) = \mathbb{E}_{p_\theta(\tilde{x}_0^i | \mathbf{x}_t)}[q(x_{t-1}^i = x_0^i | x_t^i = \texttt{[MASK]}, \tilde{x}_0^i)] = p_\theta(x_0^i | \mathbf{x}_t) q(x_{t-1}^i = x_0^i | x_t^i = \texttt{[MASK]}, x_0^i).$$

Plugging in and canceling equal factors then results in the desired expression

$$D = \sum_{i=1}^{L} -\delta_{x_t^i = \texttt{[MASK]}} \frac{\alpha_{t-1} - \alpha_t}{1 - \alpha_t} \log p_\theta(x_0^i | \mathbf{x}_t).$$

$\square$

