# OpenReview forum: "Locally Coherent Parallel Decoding in Diffusion Language Models"
_ICML.cc/2026/Conference — ICML 2026 regular_

### Official Review · Reviewer_Pdhs · 2026-02-22

**Soundness:** 3
**Presentation:** 4
**Significance:** 3
**Originality:** 3
**Overall Recommendation:** 5
**Confidence:** 5

**Summary:**

This paper tackles a core limitation of current masked diffusion language models (DLMs): during decoding, masked tokens are typically sampled independently, breaking local coherence and multi-token structures. The authors propose CoDiLA, which first partitions the sequence into fixed-size contiguous blocks, runs diffusion over blocks, and delegates within-block joint decoding to a small auxiliary auto-regressive (AR) model. The AR model is conditioned on the DLM’s per-token marginals via a prefix of expected embeddings. The paper provides theoretical justification for block-level diffusion processes with local dependency modeling, and empirically improves the accuracy-throughput Pareto frontier on code generation benchmarks.

**Compliance With Llm Reviewing Policy:**

Affirmed.

**Final Justification:**

I recommend accept. This paper proposes a practically useful method for improving local coherence in masked diffusion language models through block-wise diffusion with a lightweight autoregressive decoder. I found the work technically sound and practically relevant, with convincing gains on the targeted accuracy-throughput tradeoff. My main concerns were well addressed in the rebuttal through additional experiments and clarifications, which reinforced my original positive assessment. After reading the authors’ response and the other reviews, I remain confident in my recommendation.

**Key Questions For Authors:**

- Results are shown only for code generation benchmarks, and it is unclear how well CoDiLA transfers to other settings, such as general text generation, infilling tasks, and math reasoning, where local structure and long-range constraints may differ.
- How is the AR readout head trained in CoDiLA? Does the training implementation require duplicating context across blocks, similarly to typical block diffusion models (as Figure 2 might suggest)?
- Did you explore and ablate alternative ways to inject DLM marginals into the AR model beyond the expected embedding prefix (e.g., simply adding the marginal embedding at each position, with or without thinking tags, etc.)?
- What happens if the within-block model is purely AR without any marginals from the frozen DLM? This means a full hybrid model with globally diffusion processes and locally autoregression. How do you interpret the resulting probabilistic formulation, and how does the performance/throughput compare?
- How small can the AR decoder be before performance degrades substantially? A scaling study (model size versus performance or throughput) might help clarify how much capacity is needed to capture local dependencies effectively.

**Limitations:**

The authors have adequately discussed the limitations and potential negative societal impact of their work.

**Strengths And Weaknesses:**

Strengths:
- Clear motivation and problem framing. The conditional-independence issue in current DLMs is important, well articulated, and supported by intuitive examples.
- The paper is well written and easy to follow, with clear formulation and description.
- The hybrid design (block diffusion with a local small AR decoder) is pragmatic and emphasizes the throughput-accuracy tradeoff, aligning with the main bottleneck for current diffusion language models.

Weaknesses:

- Throughput comparison may be affected by implementation details (framework, KV caching, batching), especially with hybrid designs. In particular, the local AR component may further complicate throughput comparisons. Reporting results beyond batch size 1 would strengthen the key Pareto-frontier claim.
- While this approach is an effective compromise, block-wise left-to-right decoding can implicitly bias generation order and may introduce block boundary issues. The paper would benefit from a more explicit analysis or discussion of boundary-related errors and any-order behavior.

---

> ### Author Rebuttal · Authors · 2026-03-31
>
> We thank the reviewer for their thoughtful and very positive review. Particularly, we are glad that they appreciate our clear motivation and problem framing, our clear formulation and writing, and the efficiency of our hybrid design.
>
> Motivated by the review, we now include throughput comparisons with larger batch sizes, demonstrate any-order behavior through generation analysis, evaluate CoDiLA on complex graph traversal and code infilling tasks (which necessitate bidirectional modeling), show the benefits of our soft conditioning interface, and ablate the robustness of the AR model size.
>
> ---
>
> ## W1: Results Beyond Batch-size 1
> We agree that in contrast to the DLM backbone, our small local AR cannot fully utilize the GPU in batch size 1 settings. As a result, moving to larger batches implies significantly smaller slowdown compared to the Dream-Coder-Instruct-7B baseline, and thus larger margins in our Pareto-front optimality. We show the throughput (in tokens/s on A100-80GB) in the table below for K=B=8 on HumanEval:
>
> | Batch Size | Dream-Coder-7B | CoDiLA |
> |-|-|-|
> | 1  | 74.6   | 61.4   |
> | 2  | 74.4   | 66.5   |
> | 4  | 72.6   | 70.4   |
> | 8  | 71.0   | 72.8   |
>
> As we can see, the computational cost of the AR model is effectively marginalized at larger batch sizes.
>
> ---
>
> ## W2: Block Boundary Issues and Any-order Behavior
> CoDiLA could benefit from adaptive block sizes to ensure semantic boundaries. This might be particularly helpful in coding, where each line could be represented by a block. We envision future work to dynamically select blocks based on techniques such as Swordsman (Zhang et al. 2026), or even allowing the AR model to either expand or reduce the block size, similar to DreamOn (Wu et al. 2026).
>
> To analyze the any-order generation capabilities of CoDiLA, we compute the Spearman correlation between AR generation and CoDiLA's sampling order on MBPP. Interestingly, correlation decreases as block size increases (i.e., more any-order behavior), indicating that by delegating local dependencies to the AR, CoDiLA preserves (and even improves) the global any-order advantages of the underlying DLM.
>
> | B/K | Dream-Coder-7B | CoDiLA   |
> |-|-|-|
> | 2   | 0.9998 | 0.9186   |
> | 4   | 0.9999 | 0.9021   |
> | 8   | 0.9994 | 0.8344   |
>
> ---
>
> ## Q1: Additional Graph Traversal and Infilling Tasks
> We extended our evaluation to code infilling and synthetic graph traversal, demonstrating CoDiLA's ability to handle bidirectional dependencies that challenge standard AR models. These results show that CoDiLA improves local coherence and accelerates generation in highly parallel decoding regimes, as detailed in the response to W2 of `BaPL`.
>
> ---
>
> ## Q2: Training the AR readout
> The AR readout head is trained via cross-entropy loss on a fully-masked block determined by the forward diffusion process. Note that this is consistent with Eq. (2) except that each token is replaced by a block of tokens. As a result, context does not have to be duplicated across blocks. The input to the AR model directly comes from the DLM predictions, and the output is given by the ground-truth tokens of the masked block.
>
> ---
>
> ## Q3: Ablating Soft-Conditioning Interfaces
> Following your suggestion, we contrast the NELBO achieved with `<think>` `</think>` delimiting tokens against the one achieved without them for blocks of size $4$ after training convergence. We find that the delimiters yield a lower loss (13.6), compared to the interface without them (15.5).
>
> Moreover, we extended our ablation on the top-k tokens used for soft-conditioning of the AR model in CoDiLA (see response to W3 of `S5ow`).
>
> ---
>
> ## Q4: AR Without Conditioning
> We are not sure if we understood the question correctly. If the within block model was purely AR without any conditioning on the DLM, it would predict random (unconditioned) tokens and performance would drop to zero. If we condition it (in-context) on the DLM predicted top-1 tokens, we get the performance indicated in Table 1. If the AR model is run directly on the DLM predicted top-1 tokens without in-context conditioning, next-token prediction would only be able to ensure single-step coherence and would not scale to larger block sizes.
>
> ---
>
> ## Q5: Size of AR Decoder
> We have now conducted an ablation on the AR model size and can indeed confirm that its size does not have any consistent impact on downstream performance (see response to W1 of `BaPL`).
>
> ---
>
> Once again, we thank the reviewer for their detailed feedback. In light of our clarifications and additional results we hope to have reinforced their decision to recommend acceptance.
>
> Zhang et al. (2026). Swordsman: Entropy-Driven Adaptive Block Partition for Efficient Diffusion Language Models. arXiv:2602.04399.
>
> Wu et al. (2025) Diffusion Language Models for Code Infilling Beyond Fixed-size Canvas. ICLR.
>
> Ye et al. (2025) Beyond Autoregression: Discrete Diffusion for Complex Reasoning and Planning. ICLR.

---

> > ### Author Rebuttal · Reviewer_Pdhs · 2026-04-01
> >
> > Thank the authors for the detailed clarifications and additional experiments. The added batch-size throughput results, any-order analysis, and extended evaluations address my main concerns. Regarding “AR without conditioning,” I meant keeping the block-level factorization and modeling the joint of tokens within each block autoregressively without explicit conditioning on diffusion outputs (since the transformer layer can still capture cross-block dependencies implicitly via attention); your explanation and added ablations sufficiently address this point. Overall, my concerns are adequately resolved, and I maintain my original score. Nice work, and good luck.

---

> > > ### Author Response · Authors · 2026-04-08
> > >
> > > We sincerely thank the reviewer for their strong support and for the insightful suggestions provided during the review process. Their prompts to investigate larger batch-size throughput and any-order behavior were particularly valuable; they allowed us to demonstrate that CoDiLA’s advantages are even more pronounced in practical production-scale settings. We also appreciate the clarification regarding "AR without conditioning". Exploring block-wise factorization with purely implicit attention-based dependencies is indeed an intriguing direction for future work to optimize computational overhead. We are glad our additional experiments addressed the remaining concerns, and we thank the reviewer for helping us improve the impact of our work.

---

### Official Review · Reviewer_4rFo · 2026-03-03

**Soundness:** 3
**Presentation:** 3
**Significance:** 3
**Originality:** 3
**Overall Recommendation:** 4
**Confidence:** 4

**Summary:**

Diffusion language models (DLMs) offer sub-linear latency and bidirectional capabilities but suffer from incoherent parallel token generation due to conditional token independence bias. This paper proposes CoDiLA, a hybrid framework that delegates local coherence to a compact auxiliary autoregressive (AR) model soft-conditioned on DLM’s marginal distributions, while retaining DLM’s core strengths via block-level diffusion modeling. Overall, a pertinent challenge considered by this study is reconciling DLM’s parallel efficiency with local syntactic coherence without excessive overhead or lost capabilities. Experiments on code benchmarks (HumanEval, MBPP, BigCodeBench) show CoDiLA achieves a new Pareto frontier for accuracy and throughput, with >2× speedup and comparable accuracy via dynamic parallelism. Overall, the paper's primary contribution comprises block-level diffusion modeling, a soft-conditioning interface for DLM-AR integration, and confidence-based unmasking to balance efficiency and coherence.

**Compliance With Llm Reviewing Policy:**

Affirmed.

**Final Justification:**

Thanks for the rebuttal. My concern has been fully addressed and I recommend to accept this paper.

**Key Questions For Authors:**

NA

**Limitations:**

The tokenizer should be compatible with ar model and dllm.

**Strengths And Weaknesses:**

# Strengths
1. Innovatively proposed a hybrid framework combining block-level diffusion modeling with a lightweight auxiliary AR model. Theoretically, it is proven that intra-block joint likelihood modeling can strictly reduce NELBO compared to traditional token-level independent modeling. Meanwhile, by limiting the scope of action of the AR model, the sub-linear latency advantage of DLM is retained, effectively reconciling the core contradiction between parallel efficiency and local consistency.
2. The designed soft conditioning interface solves the adaptation problem between DLM and AR models. It theoretically verifies the necessity and sufficiency of soft embedding of the full marginal distribution, avoids the irreducible bias caused by Top-k hard truncation, and enables a compact AR model (0.6B parameters) to adapt to DLM with only a small amount of fine-tuning, taking into account both performance and deployment efficiency.
3. The experimental design is comprehensive and targeted, covering both static and dynamic parallel demasking strategies. The effectiveness of the method is verified on various code generation benchmarks (including enhanced versions) such as HumanEval, MBPP, and BigCodeBench. Through comparison with competitors like ADJUST, it clearly demonstrates advantages in the accuracy-throughput Pareto frontier.
4. The method has good compatibility and practicality. It can be combined with existing DLM acceleration frameworks such as KV caching and block diffusion, and can be implemented by only fine-tuning the auxiliary AR model without modifying the DLM backbone network. The dynamic parallel strategy can also adaptively adjust the demasking scale according to confidence, balancing speed and accuracy.
# Weaknesses
1. The selection of block size relies on empirical tuning. Although theoretically, the larger the block size, the smaller the modeling error, in practice, limited by inference latency, only the case of B≤8 is verified. The performance boundary and optimization scheme for larger block sizes are not explored, and there is a lack of a dynamic adjustment mechanism for adaptive block sizes.
2. Although limiting the dynamic candidate range to 10 blocks can avoid premature EOS prediction, it is essentially a heuristic strategy. Expanding the range will lead to a decrease in accuracy, and no combination scheme with methods such as EOS penalization is provided, which limits the flexibility of full-sequence decoding.
3. It only focuses on code generation tasks and has not verified its generalization ability in other structured or unstructured tasks such as natural language generation. Moreover, the adaptation of the auxiliary AR model depends on the tokenizer compatibility between DLM and AR models, and adapting models with different vocabularies may require additional token mapping work.

---

> ### Author Rebuttal · Authors · 2026-03-31
>
> We thank the reviewer for their thoughtful review. Particularly, we are glad that they appreciate our theoretically motivated hybrid framework that retains sub-linear latency, our necessary and sufficient soft-embeddings as an interface between DLM and (compact) AR models, our comprehensive and targeted experimental design, and the compatibility of CoDiLA with other acceleration frameworks and methods.
>
> Thanks to the review, we further investigated the impact of the block size, came up with a global block selection strategy (i.e., global candidate scope), and extended the experiments with a more challenging code infilling as well as a graph traversal task.
>
> In the following, we address each of the concerns raised.
>
> ---
>
> ## W1: Block Size Ablation
>
> Thanks for raising this crucial point in CoDiLA. Indeed, the larger the block size, the smaller the modeling error. To strengthen this claim further and to verify it on $B>8$, we have extended Figure 3 with block sizes up to 32. We now get the following NELBO upon training convergence:
> - block size 2: 53.1%
> - block size 4: 40.8%
> - block size 8: 31.4%
> - block size 16: 24.4%
> - block size 32:  19.0%
>
> As correctly noted by the reviewer, despite the smaller modeling error, larger blocks defeat the latency benefits of parallel decoding, hence our focus on $B\leq 8$.
>
> In general, CoDiLA could indeed benefit from adaptive block sizes to ensure semantic boundaries. This might be particularly helpful in coding, where each line could correspond to a block. We envision future work to dynamically select blocks based on techniques such as Swordsman (Zhang et al. 2026), or even allowing the AR model to generate either more or less tokens than in the given context, effectively implementing expansion/deletion operations similar to DreamOn (Wu et al. 2026).
>
> ---
>
> ## W2: Global Candidate Scope
> Thank you for bringing this up. We have experimented on a variant of CoDiLA that removes the 10-block scope by instead adaptively selecting the 10 blocks of smallest DLM entropy at arbitrary global positions, followed by CoDiLA as described in the submission. We are happy to report that under an appropriate EOS penalty, CoDiLA does not require the initially proposed locally limited scope, i.e., can operate fully globally. Accordingly, we now report these results across all benchmarks. For instance, on MBPP, we obtain the following performance:
>
> | Method |     K = 1    |   B = K = 2  |   B = K = 4  |   B = K = 8  |
> |-----------------------|--------------|--------------|--------------|--------------|
> | Dream-Coder-7B        | 74.4% / 9  | 54.9% / 19 | 24.4% / 37 |  08.5% / 74 |
> | CoDiLA (local scope)    |  N/A / N/A | 68.9% / 27 | 55.6% / 47 | 39.3% / 81 |
> | CoDiLA (global scope) |  N/A / N/A | 66.9% / 24 | 55.6% / 46 | 43.2% / 87 |
> *Note: Results are reported as Pass@1 / TPS.*
>
> The same trends also hold for the other tested benchmarks.
>
> ---
>
> ## W3-a: Tasks Beyond Coding
> We extended our evaluation to code infilling and graph traversal, demonstrating CoDiLA's ability to handle bidirectional dependencies that challenge standard AR models. These results show that CoDiLA improves local coherence and accelerates generation in highly parallel decoding regimes, as detailed by the results shown in the response to W2 of `BaPL`. We agree that extending CoDiLA to other structured/unstructured tasks such as natural language generation represent exciting opportunities to extend CoDiLA.
>
> ---
>
> ## W3-b: Compatibility Between Tokenizers
> We see the tokenizer matching requirement not as a structural limitation of CoDiLA, but rather a benefit of our "minimal adaptation" design. For instance, LLaDA could be seamlessly integrated with a compact Llama-2-based AR model (e.g., TinyLlama). In cases where a pre-trained match is unavailable, the auxiliary AR head is sufficiently lightweight to be trained from scratch on the target vocabulary, as demonstrated in our additional graph traversal task results (see response to W2 of `BaPL`).
>
> ---
>
> Once again, we thank the reviewer for their detailed feedback. In light of our clarifications and additional evidence we hope to have reinforced their decision to recommend acceptance.
>
> Ye et al. (2025) Beyond Autoregression: Discrete Diffusion for Complex Reasoning and Planning. ICLR.
>
> Zhang et al. (2026). Swordsman: Entropy-Driven Adaptive Block Partition for Efficient Diffusion Language Models. arXiv:2602.04399.
>
> Wu et al. (2025) Diffusion Language Models for Code Infilling Beyond Fixed-size Canvas. ICLR.

---

> > ### Author Rebuttal · Reviewer_4rFo · 2026-04-02
> >
> > My concerns have been adequately addressed. I will keep my score. Thanks for the detailed rebuttal.

---

> > > ### Author Response · Authors · 2026-04-08
> > >
> > > We thank the reviewer for their time and thoughtful evaluation of our work. Their feedback was instrumental in pushing us to explore larger block size ablations and the global candidate scope, which have significantly strengthened the paper's empirical foundation. We are pleased that these additional results fully addressed their concerns, and we appreciate their continued support for the acceptance of our work.

---

### Official Review · Reviewer_BaPL · 2026-03-12

**Soundness:** 3
**Presentation:** 3
**Significance:** 2
**Originality:** 2
**Overall Recommendation:** 3
**Confidence:** 3

**Summary:**

This paper addresses the local incoherence problem in discrete diffusion language models (DLMs), where parallel token sampling from independent conditional marginals produces sequences that are individually plausible but jointly inconsistent, which is particularly damaging for structured outputs like code. The authors propose CoDiLA, which partitions the generated sequence into fixed-size blocks and delegates within-block decoding to a lightweight auxiliary autoregressive model (Qwen3-0.6B) soft-conditioned on the full marginal distributions from a frozen DLM backbone (Dream-Coder-Instruct-7B). Overall, the paper's primary contribution comprises a theoretically grounded hybrid framework showing that block-wise factorization strictly reduces the irreducible NELBO compared to token-wise independence, and that conditioning on full marginals rather than top-k truncations is necessary to avoid irrecoverably excluding high-probability sequences, motivating a soft-conditioning interface that projects DLM marginals into the AR embedding space as expected embeddings. Overall, a pertinent challenge considered by this study is that existing coherence-enforcement methods either sacrifice the DLM's unique non-causal capabilities by forcing left-to-right generation or incur heavy computational overhead through global auxiliary models, and CoDiLA addresses this by confining autoregressive computation to short bounded blocks with static and dynamic confidence-based unmasking, establishing a new Pareto frontier on six code-generation benchmarks while preserving sub-linear latency.

**Compliance With Llm Reviewing Policy:**

Affirmed.

**Final Justification:**

The rebuttal partially addressed my concerns. However, my concern about the practical applicability of the proposed method remains. It is unclear in what real-world scenarios this approach would be preferred over existing alternatives. I maintain my original rating.

**Key Questions For Authors:**

The paper motivates CoDiLA by arguing that DLMs offer unique non-causal capabilities worth preserving. However, CoDiLA introduces an AR component precisely to compensate for a fundamental weakness of DLMs (local incoherence). Could the authors clarify the intended deployment scenario? In standard code generation (the only setting evaluated), AR models are both more accurate and more mature. For non-causal tasks where DLMs should shine, no results are provided. It is unclear what practical setting would justify the complexity of running both a 7B DLM and a 0.6B AR model, rather than simply using an AR model directly or a DLM with more denoising steps to reduce incoherence.

**Limitations:**

Yes

**Strengths And Weaknesses:**

Strengths:

Clean theoretical framework with practical impact. The paper provides a well-structured theoretical foundation that directly informs design choices. The generalization of the NELBO analysis from token-level to block-level factorization cleanly quantifies the benefit of joint modeling via within-block total correlation, and the Fréchet-class argument rigorously justifies why soft-conditioning over full marginals is necessary rather than merely preferable. Importantly, these are not just theoretical curiosities: both results translate into concrete architectural decisions (block-based decoding and soft-conditioning interface), and the ablation experiments confirm the predicted effects empirically.


Weakness:
1. Missing comparison with standalone AR baselines. The paper uses Qwen3-0.6B purely as an auxiliary component but never reports its independent code generation performance. If Qwen3-0.6B alone achieves comparable Pass@1 to CoDiLA (e.g., ~50-60% on HumanEval), the added complexity of running a 7B DLM plus soft-conditioning becomes hard to justify.

2. No evaluation of the claimed non-causal capabilities. The paper's central architectural argument is that CoDiLA preserves the DLM's unique non-causal capabilities (infilling, self-correction, bidirectional editing), explicitly criticizing left-to-right methods for sacrificing them. Yet all six benchmarks are standard prompt-to-completion tasks. Structured constraint-satisfaction tasks like Sudoku would be particularly valuable here, as they demand exactly the bidirectional cross-block correction CoDiLA claims to retain and have precisely verifiable solutions. Relevant evaluation settings like: PRISM (Kim et al., 2025) tests self-correction on Sudoku, the Code Revision Benchmark (Zhang et al., 2025b) evaluates in-place error correction, ParallelBench (Kang et al., 2025) targets tasks with strong token dependencies, and ReMDM (Wang et al., 2025a) demonstrates infilling via remasking. Without testing on any such scenario, the key advantage claimed over simpler left-to-right alternatives remains unsupported.

---

> ### Author Rebuttal · Authors · 2026-03-31
>
> We thank the reviewer for their thoughtful feedback. Particularly, we are glad that they appreciate our 'clean theoretical framework with practical impact', and recognize that our theoretical results 'translate into concrete architectural decisions' whose 'predicted effects are confirmed empirically'. In the following, we address each of the concerns raised.
>
> Motivated by their suggestions, we have included a standalone AR baseline and extended our experimental evaluations with tasks specifically designed to test non-causal capabilities, namely code infilling and graph-based planning.
>
> ---
>
> ## W1: Comparison to AR Baseline
> Thanks for raising this point, the initial submission was indeed missing results on Qwen3-0.6B alone. We now include results with an isolated AR model with the following Pass@1 scores:
> - HumanEval: 35%
> - HumanEval+: 32%
> - MBPP: 19%
> - MBPP+: 19%
>
> CoDiLA with $B\leq4$ outperforms Qwen3-0.6B across all benchmarks. Hence, CoDiLA's performance does not rely on a strong AR backbone. Rather, the AR acts as a parameterized n-gram model that reads out coherent solutions without requiring strong standalone (coding) capabilities.
>
> Our new scaling ablation on MBPP further corroborates this, showing that larger AR models provide no consistent accuracy gain, as the DLM backbone remains the primary source of task-specific knowledge:
>
> | Method (AR model) | K = 1 | B = K = 2 | B = K = 4 | B = K = 8 |
> |-|-|-|-|-|
> | Dream-Coder-7B      | 74.4% / 9 | 54.9% / 19 | 24.4% / 37 | 08.5% / 74 |
> | CoDiLA (Qwen3-0.6B) | N/A / N/A | 68.9% / 27 | 55.6% / 47 | 39.3% / 81 |
> | CoDiLA (Qwen3-1.7B) | N/A / N/A | 66.5% / 26 | 54.5% / 46 | 37.0% / 75 |
> | CoDiLA (Qwen3-4.0B) | N/A / N/A | 72.0% / 24 | 55.6% / 40 | 36.2% / 62 |
> *Note: Results are reported as Pass@1 / TPS.*
>
> The same trend holds also for the other benchmarks. Further decreasing the model size through, e.g., layer pruning, represents exciting future avenues.
>
> ---
>
> ## W2: Non-causal Capabilites
>
> Thank you for bringing this up. We have added two tasks that explicitly challenge AR models due to their bidirectional modeling requirements.
>
> **1. Code Infilling** We now include an infilling task by implementing CoDiLA on top of DreamOn (Wu et al. 2026) for HumanEval-Infilling (multi-line). Note, that this task is particularly challenging for AR models despite being trained with specific infilling templates. Using CoDiLA with dynamic parallelism (B=4), we are able to successfully accelerate the generation while maintaining a high accuracy, particularly outperforming related AR models:
>
> | Method                   | Accuracy (%) | Token/iteration |
> |-|-|-|
> | Deepseek-Coder-6.7B      | 45.7         | 1  |
> | Seed-Coder-8B            | 59.3         | 1  |
> | Qwen2.5-Coder-7B         | 58.7         | 1  |
> | DreamOn K=1 (repr.)      | 62.5         | 1  |
> | DreamOn K=2 (repr.)      | 53.1         | 2  |
> | DreamOn+CoDiLA (tau=0.2) | 62.5        | 1.3 |
> | DreamOn+CoDiLA (tau=0.5) | 61.5        | 1.5 |
>
>
> **2. Graph Traversal (Planning)** We evaluate planning capabilities on the graph traversal task from Ye et al. (2025) by integrating CoDiLA with the MGDM backbone. This experiment confirms that our auxiliary AR head can be effectively trained from scratch for tasks with specialized tokenizers. To maintain compatibility with MGDM’s unmasking schedule, we utilize the AR head as a scoring function to penalize locally incoherent candidates during parallel sampling. CoDiLA consistently improves performance across highly parallel decoding regimes:
>
> | Decoding iterations | MGDM (repr.) | CoDiLA (B=4) | CoDiLA (B=8) |
> |-|-|-|-|
> | 2 | 94.4%|         95.3%|     **95.4%**|
> | 4 | 96.5%|         96.8%|     **96.9%**|
> | 8 | 97.8%|     **98.2%**|     **98.2%**|
> | 16| 99.2%|     **99.3%**|     **99.3%**|
> | 20| 99.3%|     **99.5%**|         99.4%|
>
> ---
>
> ## Additional Questions
> The original intended deployment scenario for CoDiLA is a code assistant that writes and/or edits complex code, e.g., containing a class with dozens of methods. We hope that our additional results on HumanEval-Infilling better clarify the intended use-cases for CoDiLA, but note that unlocking the full potential of CoDiLA might require additional developments in the field such as effective remasking of masked DLMs and stronger base models.
>
> Finally, we remark that our additional experiments on graph traversal with MGDM demonstrate that the applicability of CoDiLA might exceed our initial coding-based motivation.
>
> ---
>
> Once again, we thank the reviewer for their detailed feedback. In light of our clarifications and additional evidence we hope to have convinced them to recommend acceptance.
>
> Wu et al. (2025) Diffusion Language Models for Code Infilling Beyond Fixed-size Canvas. ICLR.
>
> Ye et al. (2025) Beyond Autoregression: Discrete Diffusion for Complex Reasoning and Planning. ICLR.

---

> > ### Author Rebuttal · Reviewer_BaPL · 2026-04-03
> >
> > I thank the authors for their effort in providing additional experiments. The standalone AR baseline (W1) is convincing and I appreciate this addition. However, my core concern regarding the practical justification of the hybrid architecture remains only partially addressed.
> >
> > On the non-causal evaluation (W2): While I appreciate the two new experiments, neither of the above benchmarks I suggested was attempted. The chosen alternatives happen to be settings where CoDiLA performs conservatively: the infilling experiment shows no accuracy gain over the base DLM and only marginal speedup. Tasks like Sudoku (PRISM) , high-uncertainty task (ParallelBench), in-place correction (DLM) would more directly stress-test the cross-block coherence that CoDiLA claims to provide, and their absence leaves the central claim undersupported.
> >
> > On the deployment scenario: For standard code generation, which remains the only well-supported use case, the justification for maintaining two models over a single AR model of comparable total size is still missing.
> >
> > These concerns touch on the core contribution of the paper: whether the proposed hybrid framework offers a compelling advantage over simpler alternatives in any concrete setting.

---

> > > ### Author Response · Authors · 2026-04-08
> > >
> > > We thank the reviewer for the constructive dialogue. We are glad that the additional standalone AR baseline proved convincing. To address the remaining concerns regarding practical justification and non-causal evaluation, we provide further evidence and clarification below.
> > >
> > > ## Motiviation for Block Structure
> > > To directly address the concerns about cross-block coherence, it is important to reiterate the scope of CoDiLA. CoDiLA is designed to solve local incoherence within a parallel decoding step by soft-conditioning a local AR model on the DLM's marginals. It ensures that the tokens within a specific block are jointly modelled (hence maintaining correct syntax).
> > >
> > > CoDiLA's accuracy is fundamentally ceiled by the baseline DLM's sequential accuracy (K=1). CoDiLA's primary achievement is enabling parallel generation speed while maintaining DLM's sequential baseline accuracy. We want to stress that CoDiLA achieving similar accuracy as the base DLM in the code infilling tasks is exactly the desired outcome: we achieve speedups with almost no accuracy degradation.
> > >
> > > ## CoDiLA's Empirical Value
> > >
> > > **1. Practical Justification Through Infilling Experiments**
> > > Why using a hybrid 7.6B model over a single AR model? Our results on Code Infilling provide the answer: **Similar-sized AR models (7-8B) are strictly outperformed by CoDiLA (7.6B) in both accuracy and generation efficiency for infilling tasks.** While AR models can be forced to infill via prompting using particular templates, they lack the native bidirectional modeling of DLMs. CoDiLA allows a DLM to maintain this bidirectional capabilities while fixing its primary weakness (local syntax errors in parallel coding) achieving 1.5 tokens per DLM call.
> > >
> > > **2. Synthetic Tasks**
> > > While our primary focus is on practical, large-scale structured tasks where bidirectional context and strict syntax are functional requirements, we agree that synthetic tasks provide valuable stress tests.
> > >
> > > **ParallelBench (Waiting Line Task)**
> > >
> > > Prompted by the reviewer's suggestion, we tested CoDiLA on the ParallelBench waiting line tasks. We observed that standard CoDiLA achieves similar accuracy to the baseline DLM with static top-k entropy decoding. Investigations revealed that most errors generated in these tasks are semantic, not syntactic.
> > > - For example, when asked to shuffle the list \"Patrick Hicks\", \"Eric Reynolds\", \"Roy Ortiz\", the DLM might output the first entry like \"Roy Reynolds\".
> > > - The local AR model accepts this because it is syntactically correct (proper formatting, valid name sequence). CoDiLA successfully enforces the local joint probability of the text structure, but it cannot fix a global semantic constraint violation (sampling without replacement). This perfectly isolates CoDiLA's role: it is a local syntax/coherence enforcer, while global logic remains the domain of the DLM.
> > >
> > > Nevertheless, using the AR model as a verifier function to penalize locally incoherent candidates during parallel sampling (as we did in the Graph Traversal task) is a highly promising avenue. Our preliminary results indeed show benefits in highly parallel regimes (K$\geq$4).
> > >
> > > | **Task**| **K=1** | **K=2** | **K=4** | **K=8** |
> > > | -------------- | - | ------- | ------- | ------- |
> > > | Dream-Instruct (repr.) | **87.2** | **86.8** | 64.9 | 54.1 |
> > > | Standard CoDiLA |  | **86.8** | 61.1 | 53.0 |
> > > | CoDiLA Verifier | | 77.6 | **68.1** | **60.2** |
> > > _Average accuracy on waiting line in %_
> > >
> > > **Graph Traversal**
> > >
> > > As shown in our initial rebuttal (W2), CoDiLA enhances MGDM's parallel decoding performance in synthetic graph traversal (planning). Standalone AR models fail on this task due to missing bidirectional modeling
> > >
> > >
> > > **3. Other Non-Causal Tasks for Future Works**
> > > The reviewer's suggestion regarding in-place correction is a highly relevant use case for CoDiLA. By treating a block as the atomic unit of re-masking, future iterations of CoDiLA could regenerate corrupted code segments using full bidirectional context from both sides. This would theoretically achieve higher coherence than a standard parallel DLM, without the latency penalty of sequential token-by-token AR correction.
> > >
> > > ---
> > >
> > > By providing a 1.5x higher token-per-call efficiency over SoTA infilling models and outperforming standalone AR models of comparable size, CoDiLA offers a clear, practical path for deploying DLMs. We believe that our results and clarifications support the CoDiLA’s central claims and practical significance.

---

### Official Review · Reviewer_S5ow · 2026-03-13

**Soundness:** 3
**Presentation:** 3
**Significance:** 3
**Originality:** 3
**Overall Recommendation:** 4
**Confidence:** 4

**Summary:**

This paper proposes CoDiLA, a hybrid framework combining a bidirectional DLM with a small auxiliary AR model to address local incoherence in parallel decoding. The sequence is partitioned into blocks; the DLM produces token-wise marginal distributions globally, and a compact AR model (Qwen3-0.6B) decodes each block conditioned on soft embeddings from those marginals. Two theorems support the design: block factorization reduces NELBO versus token-level independence, and soft-conditioning on full marginals is necessary and sufficient for coherent recovery. Experiments on code generation benchmarks (HumanEval, MBPP, BigCodeBench) compare CoDiLA against Dream-Coder-Instruct-7B and ADJUST.

**Compliance With Llm Reviewing Policy:**

Affirmed.

**Final Justification:**

The authors have successfully addressed my main concerns and I have changed my position to weak accept from reject.

**Key Questions For Authors:**

Q1. The justification for inter-block coherence — bidirectional DLM attention and sequential block unmasking — applies equally to token-level denoising. What specific property of intra-block token dependencies requires block structure and an auxiliary AR model that cannot be resolved by the same DLM mechanisms at token granularity?

Q2. Since CoDiLA (7B + 0.6B) is never compared against a parameter-matched DLM baseline, how can the authors attribute the observed Pass@1 improvements to the block-coherence mechanism rather than the additional 0.6B parameters and their SFT finetuning? What prevents a 7.6B DLM from achieving comparable or better results?

Q3. Given that accuracy degrades monotonically with increasing block size B across all benchmarks and no CoDiLA configuration exceeds the sequential DLM ceiling, can the authors provide a setting where block modeling demonstrably improves over both the sequential DLM baseline and matched-throughput alternatives? If not, the practical motivation for the framework requires substantial revision.

**Limitations:**

Not enough comparison with DLM baselines, contradictory between theoretical hypothesis and empirical results.

**Strengths And Weaknesses:**

S1. The Fréchet class argument in Theorem 3.3 is technically sound and non-trivial, providing a principled justification for soft over hard conditioning, supported by the ablation in Table 1.

S2. The soft-conditioning interface via expected embeddings is an elegant design, enabling use of a pretrained AR model with minimal finetuning and no vocabulary remapping.

S3. Dynamic parallelism (Section 3.5) is practically valuable, demonstrating in Figure 5 that CoDiLA can maintain >2× speedup while partially recovering sequential decoding accuracy.

**W1. The block structure motivation is internally inconsistent.**
The authors justify inter-block coherence through the DLM's bidirectional attention and sequential unmasking of blocks as hard context. However, this same mechanism applies equally to token-level denoising. If global bidirectional attention suffices for inter-block dependencies, it should equally resolve intra-block token dependencies — making the block structure and auxiliary AR model unnecessary by the paper's own logic. This is a fundamental gap that undermines the core motivation.

**W2. CoDiLA never outperforms the sequential DLM baseline in absolute accuracy.**
Examining Figure 4, CoDiLA at its lowest parallelism setting (B=2) consistently underperforms Dream-Coder-Instruct-7B and ADJST at K=1 across all benchmarks. As block size increases, the accuracy degradation is substantial — suggesting that block-level masking actively hinders the DLM's performance rather than improving it. The paper's Pareto frontier claim holds only at matched throughput, not in absolute terms, which severely limits the practical significance of the contribution.

**W3. Key ablation in Table 1 lacks external baselines.**
Table 1 compares only two CoDiLA variants (soft vs. hard conditioning) against each other, without including the base DLM at sequential decoding (K=1) as a reference ceiling. While Figure 4 includes this comparison graphically, its omission from Table 1 makes the ablation difficult to interpret in isolation.

**W4. No direct empirical evidence of coherence improvement.**
Despite coherence being the central claim, all experiments use Pass@1 as the sole metric. Without syntactic validity rates or structural error rates before and after CoDiLA, it is impossible to attribute Pass@1 changes specifically to coherence improvement versus the AR model's general code generation capability from SFT finetuning.

---

> ### Author Rebuttal · Authors · 2026-03-31
>
> We thank the reviewer for their thoughtful feedback and for recognizing our non-trivial theoretical framework, our hybrid design requiring minimal adaptation, and the 2x speedup enabled by dynamic parallelism.
>
> We now extended Table 1 with more baselines, found evidence for coherence improvements through code syntax analysis, and ablated the robustness to the AR model size.
>
> ---
>
> ## W1 & Q1: Motivation for Block Structure
> Thanks for raising this very important point. DLMs model the conditional probability as the product of the marginals. While these marginals are correlated through bidirectional attention, independently *sampling* from the marginals can introduce incoherence. Indeed, DLMs address this by sampling only one token per denoising iteration (sequential decoding).
>
> This work enables locally coherent parallel generation of a block of tokens by *sampling from an AR model*. This way, CoDiLA can generate coherent *single blocks* instead of only *single tokens*. This can be seen as ensuring local syntax within a block (see syntax improvements in W4).
>
> The reviewer rightly mentions that in our experiments, we generate blocks of size $B$ sequentially to ensure inter-block (global) coherence. Nevertheless, note that by generating each block in parallel, we reduce the latency by a factor $B$ compared to fully sequential token unmasking using a DLM. So while the block structure does not guarantee global coherence, the introduced local coherence translates into meaningful latency improvements.
>
> If one wants to generate multiple blocks in parallel, we indeed have the same challenge as in standard DLMs. Yet, weak inter-block dependence could allow parallel block sampling. For example, hardware description languages could have limited dependencies between logic (compute) and register (storage) blocks. We see this as exciting avenue for future work.
>
> ---
>
> ## W2 & Q3: Accuracy Ceiling and Pareto Optimality
> Parallel decoding is a fundamental trade-off between speed and accuracy. Sequential DLM decoding (K=1) represents the accuracy ceiling because it maximizes conditional context for every token. CoDiLA’s goal is to push the Pareto frontier by providing higher accuracy than other parallel methods at the same speed.
>
> As shown in our dynamic parallelism experiments (Fig 5), CoDiLA recovers K=1 accuracy. We demonstrate that CoDiLA achieves a >2x speedup with negligible accuracy loss on HumanEval.
>
> ---
>
> ## W3: Key Ablations in Table 1
> Thanks for pointing this out. We updated Table 1 with the DLM baseline as well as additional ablations in CoDiLA's top-k on HumanEval. We note that Top-1 conditioning, rather than our soft-conditioning decreases the performance substantially, almost down to the DLM baseline under equivalent parallel decoding.
>
> |   Conditioning   |K = 1|B = K = 2|B = K = 4| B = K = 8|
> |-|-|-|-|-|
> | Dream-Coder-7B   | 74.4% / 9  | 54.9% / 19 | 24.4% / 37 | 08.5% / 74 |
> | CoDiLA (Softmax) | N/A / N/A | 68.9% / 18 | 51.8% / 33 | 29.9% / 60 |
> | CoDiLA (Top-5)   | N/A / N/A | 63.4% / 18 | 56.1% / 33 | 28.0% / 58 |
> | CoDiLA (Top-3)   | N/A / N/A | 61.0% / 18 | 50.0% / 31 | 31.7% / 59 |
> | CoDiLA (Top-1)   | N/A / N/A | 53.0% / 18 | 37.8% / 35 | 22.0% / 58 |
> *Note: Results are reported as Pass@1 / TPS.*
>
> ---
>
> ## W4: Empirical Evidence of Coherence Improvement
> This is a very good question. As a proxy for incoherence in coding, we now measure the syntax errors as the rate at which the extraction script of the eval-harness cannot extract any code. Using this metric, we find that CoDiLA significantly reduces syntax error rate:
>
> | Model  | K = B = 2 | K = B = 4 | K = B = 8 |
> |-|-|-|-|
> | Dream-Coder-7B    | 18%       | 38%       | 70%       |
> | CoDiLA            | 4%        | 13%       | 16%       |
>
>  Moreover, as shown in the table below, decoding (in parallel) single blocks of size 8 is more accurate than 2 blocks of size 4 (8% delta). Similarly, 1 block of size 4 outperforms two blocks of size 2 by 2.3%. This demonstrates that the local coherence achieved by our block structure is an essential ingredient for retaining performance in parallel decoding.
>
> | CoDiLA parallelism |   B = K = 2  |   B = K = 4  |   B = K = 8  |
> |-|-|-|-|
> | CoDiLA (1 block)   | 68.9% / 27 | 55.6% / 47 | 39.3% / 81 |
> | CoDiLA (2 blocks)  | 53.3% / 54 | 31.2% / 95 | 22.3% / 159 |
> *Note: Results are reported as Pass@1 / TPS.*
>
>
> ---
>
> ## Q2: Source of Improvements
> We attribute the Pass@1 improvements to block coherence rather than the additional 0.6B parameters. This is shown by additional AR scaling analysis (see response to W1 of `BaPL`), where larger AR models do not improve performance. Note that the AR model Qwen3-0.6B itself is outperformed by CoDiLA. Unfortunately, training a dedicated 7.6B DLM from scratch is computationally prohibitive.
>
> ---
>
> Once again, we thank the reviewer for their detailed feedback. In light of our clarifications and additional evidence, we hope to have convinced them to recommend acceptance.

---

> > ### Author Rebuttal · Reviewer_S5ow · 2026-04-01
> >
> > The initial score of this paper was graded low, due to my misunderstading of basic concepts of denoising steps of diffusion language models. Since, the rebuttal of the authors have successfully addressed my major concerns I have changed my score to weak accept from reject. Good luck for rest of your rebuttals.

---

> > > ### Author Response · Authors · 2026-04-08
> > >
> > > We thank the reviewer for the careful re-evaluation of our work. We are pleased that our rebuttal and the additional evidence successfully clarified the concerns regarding the denoising mechanisms and the motivation for our block structure. The reviewer's feedback significantly helped us to improve the clarity of the paper, and we appreciate the positive change in their recommendation.

---

### Decision · Program_Chairs · 2026-04-30

**Decision:**

Accept (regular)

**Comment:**

This paper proposes CoDiLA, a hybrid diffusion–autoregressive framework that addresses the local coherence failures of parallel discrete diffusion decoding. To preserve diffusion’s global bidirectional and parallel generation advantages across blocks, while using a small auxiliary autoregressive model to decode tokens sequentially within each block, the paper argues that moving from token-level independence to block factorization reduces the NELBO, and further shows that coherent recovery is achieved only when the auxiliary model is soft-conditioned on the full marginal distributions rather than hard token choices.

The paper’s problem framing is clear and important. The theoretical framework, particularly the Fréchet-class argument and the justification for soft conditioning over hard conditioning are technically sound and nontrivial. The soft-conditioning interface is also elegant and practical, which can be instantiated with a pretrained auxiliary AR model with relatively little adaptation. Empirically, the paper demonstrates a strong throughput–accuracy tradeoff on coding tasks.

The concerns regarding this paper are 1) core motivation and architectural logic; 2) no clear improvements over sequential DLM baseline in absolute accuracy; 3) lack of direct evidence for the claimed coherence improvements + baselines not complete and 4) practical issues such as throughput measurement and implementation dependence. During rebuttal, these points are addressed. The reviewers vote accept and the authors are expected to incorporate the comments into the camera-ready.